

# 2-Group symmetries and their classification in 6d

Fabio Apruzzi[1,2], Lakshya Bhardwaj[1], Dewi S. W. Gould[1] and Sakura Schäfer-Nameki[1]

**1** Mathematical Institute, University of Oxford,
Andrew-Wiles Building, Woodstock Road, Oxford, OX2 6GG, UK
**2** Albert Einstein Center for Fundamental Physics, Institute for Theoretical Physics,
University of Bern, Sidlerstrasse 5, CH-3012 Bern, Switzerland

## Abstract

We uncover 2-group symmetries in 6d superconformal field theories. These symmetries arise when the discrete 1-form symmetry and continuous flavor symmetry group of a theory mix with each other. We classify all 6d superconformal field theories with such 2-group symmetries. The approach taken in 6d is applicable more generally, with minor modifications to include dimension specific operators (such as instantons in 5d and monopoles in 3d), and we provide a discussion of the dimension-independent aspects of the analysis. We include an ancillary `mathematica` code for computing 2-group symmetries, once the dimension specific input is provided. We also discuss a mixed 't Hooft anomaly between discrete 0-form and 1-form symmetries in 6d.

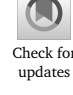
doi:10.21468/SciPostPhys.12.3.098

# 1  Introduction

Global symmetries are an essential characteristic of quantum field theories (QFTs). They determine the charges of operators – local or extended – and their 't Hooft anomalies provide an RG-flow invariant. Higher-form symmetries [1] and higher-group structures [2] are the newest members of the family of global symmetries in QFTs[1]. In the past year it has become clear that their role extends beyond low-dimensional theories, and provides a characterization of higher-dimensional QFTs, in particular superconformal field theories (SCFTs). This is particularly important in the context of strongly coupled SCFTs in 5d and 6d, whose existence is largely argued based on their string theoretic realizations. Global 0-form flavor symmetries are e.g. an important datum to test our understanding of the UV fixed points in 5d – starting with the work in [25–27] and more recently in [20, 28–73].

There is a clear distinction between higher-form symmetries that are continuous and those that are discrete. *Continuous* 1-form symmetries were shown to not exist in 5d SCFTs in [74], because of the absence of a conserved 2-form current in the spectrum of short multiplets, which are representations of the superconformal algebra. In contrast however, *discrete* 1-form symmetries and their gauged versions, 2-form symmetries, are numerous in 5d SCFTs [20, 62, 68, 75]. Moreover, in [20] it was shown that there are also 2-group symmetries in 5d SCFTs involving these discrete 1-form symmetries.

Likewise, based on the superconformal symmetry, it was shown in [15] that *continuous* 1-form symmetries, and 2-groups having *continuous* 1-form and 0-form components, cannot exist in 6d SCFTs. The presence of *discrete* 1-form symmetries in 6d SCFTs was already pointed out in [62, 68] [2]. In the present paper we show that there are 2-group symmetries in 6d SCFTs, based on *discrete* 1-form symmetries and continuous 0-form symmetries. With the full

---

[1]A wide array of recent work has been devoted to higher-group structures in QFTs. See [2–24].
[2]There is also a defect group associated to 2d charged objects in 6d [76].

classification of 6d SCFTs in place we are able to give a complete list of 6d SCFTs exhibiting a particular kind of 2-group symmetry: these are always a combination of the 1-form symmetry that is a discrete group, and the flavor symmetry group, which is a non-simply-connected continuous group. All models with 2-group symmetries in 6d are summarized in table 1, modulo the information about the possible choices of gauge groups, which can be found in section 3.3.2. In contrast, we find that this kind of discrete 2-group does not exist in little string theories (LSTs). However there definitely are continuous ones, as was shown in [15, 16].

A detailed derivation of 2-group structures will be provided in this paper. We should here give some intuition,[3] when a 2-group symmetry (of the kind studied in this paper) can be expected in a given QFT. Necessary conditions are the existence of a discrete 1-form symmetry $\Gamma^{(1)}$, but also a non-trivial 0-form flavor symmetry group, which is not simply-connected. We can write the latter as $\mathcal{F} = F/\mathcal{Z}$, where $F$ is a cover of the flavor symmetry group, and $\mathcal{Z}$ a subgroup of the center $Z_F$ of $F$. Whenever there are local operators that are charged under both flavor and gauge symmetry, these can lead to a non-trivial extension of $\Gamma^{(1)}$ by $\mathcal{Z}$. This maximal, trivially acting group will be denoted by $\mathcal{E}$ in the following. Whenever this forms a non-trivial extension

$$1 \to \Gamma^{(1)} \to \mathcal{E} \to \mathcal{Z} \to 1 \,, \tag{1.1}$$

with a non-trivial Bockstein map

$$\text{Bock}: \qquad H^2(B\mathcal{F}, \mathcal{Z}) \to H^3(B\mathcal{F}, \Gamma^{(1)}) \,, \tag{1.2}$$

where $B\mathcal{F}$ is the classifying space for the flavor symmetry $\mathcal{F}$-bundles, then there is a 2-group symmetry. The background $B_2 \in C^2(B\mathcal{F}, \Gamma^{(1)})$ is then related to the Stiefel-Whitney class $w_2 \in H^2(B\mathcal{F}, \mathcal{Z})$, that measures the obstruction of lifting $\mathcal{F}$-bundles to $F$-bundles, by

$$\delta B_2 = \text{Bock}(w_2) \,. \tag{1.3}$$

This relatively simple argument percolates throughout QFTs in all dimensions. The central theory-dependent information is the sequence (1.1), i.e. the 1-form symmetry $\Gamma^{(1)}$, the subgroup of the flavor center $\mathcal{Z}$, which acts trivially, and perhaps most importantly, the extension sequence they form. This sequence is encoded in the symmetries (gauge and flavor), matter and – depending on dimension – non-perturbative states such as instantons, monopoles and vortices etc.

In view of this general nature of the 2-group construction, we provide a computational tool in the form of a `mathematica` notebook, `TwoGroupCalculator.nb`. It simply requires the input of the symmetry groups, and charges of states (perturbative and non-perturbative alike), and outputs the 1-form symmetry and $\mathcal{E}$, as well as the embedding of the former into the latter. This specifies whether there is a non-trivial extension. Subsequently of course one still needs to determine whether $\text{Bock}(w_2)$ is a non-zero element or not. The code also specifies $\mathcal{Z}$ as a subgroup of $Z_F$, from which we then also can infer the global form $\mathcal{F}$ of the flavor symmetry group.

The non-perturbative BPS strings play a crucial role for the 1-form symmetries, as well as for the 2-groups. They give rise to massive states at low-energy but massless in the UV where the SCFTs or LSTs live. These states can sometimes screen Wilson lines transforming in the center of a gauge group, therefore breaking the 1-form symmetry. They can also mediate between gauge and flavor Wilson lines. We describe a method that, without knowing explicitly the representation of the BPS string charges, provides a consistency condition for turning on

---

[3]Here we provide this intuition by staying within the context of gauge theories, though the formalism we discuss can be applied even when no useful gauge theory description is available. For example, see [21], where this formalism was used to deduce 2-group symmetries of 4d $\mathcal{N} = 2$ theories of Class S, which generically do not admit a gauge theory description.

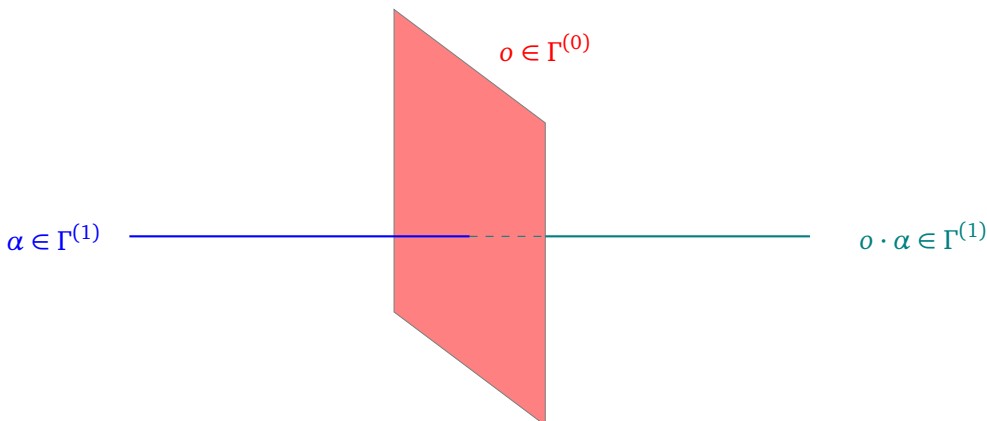

Figure 1: Action of a topological operator associated to 0-form symmetry on topological operators associated to 1-form symmetries.

the 1-form symmetry [77], 0-form flavor symmetry and/or 2-group backgrounds. This methods relies on studying the Dirac quantization of the BPS string charge lattice in the presence of these backgrounds, via the Green-Schwarz-West-Sagnotti 6d topological couplings.

Finally, we discuss also the role of abelian flavor symmetries. In particular, the classical $U(1)$ symmetries get broken by Adler–Bell–Jackiw (ABJ) anomalies at the quantum level. These anomalies sometimes leave a remnant discrete 0-form symmetry. We show that for certain 6d theories on 6-manifold with non-vanishing first Pontryagin class there is a mixed 't Hooft anomaly between the discrete 0-form symmetry and the 1-form symmetry. The existence of the 2-group is consistent with this mixed anomaly as discussed at the end of section 4.2.

The plan of this paper is as follows: in section 2 we discuss the general framework for 2-groups. This section is to a large extent dimension independent. In section 3 we apply this to the 6d SCFTs and LSTs and provide in section 5 the complete classification of theories exhibiting 2-groups (of a particular type). In section 4 we discuss anomalies – in particular the ABJ anomaly for theories with $U(1)$ global symmetries and a new mixed anomaly in 6d. The appendices supply the details of the mathematica code and a selection of detailed examples of 2-groups in 6d.

## 2 2-Group Symmetries

The construction of 2-groups has many facets. Here we describe a construction which applies to theories in general dimension $d$. What remains dimension specific is the specific type of gauge, flavor symmetry groups and charged matter states (including non-perturbative states such as strings, instantons and monopoles).

### 2.1 The Types of 2-Group Symmetries

2-group symmetries describe mixings between 0-form and 1-form symmetries of a theory. The most straightforward way this can happen is if elements of a 0-form symmetry group $\Gamma^{(0)}$ act on elements of the 1-form symmetry group $\Gamma^{(1)}$ (see figure 1). An example of such a situation arises in a pure $SU(N)$ gauge theory, which has a $\mathbb{Z}_N^{(1)}$ 1-form symmetry group that is acted upon by the $\mathbb{Z}_2^{(0)}$ charge conjugation symmetry. The action sends an element in $\mathbb{Z}_N^{(1)}$ to its inverse element.

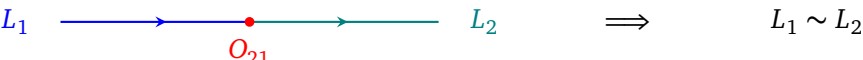

Figure 2: If there exists a non-genuine local operator $O_{21} \neq 0$ that can be used to transform a line defect $L_1$ to line defect $L_2$, then we regard $L_1$ and $L_2$ to be in the same equivalence class. Such equivalence classes form a group under OPE of line defects, which can be recognized as the Pontryagin dual group $\widehat{\Gamma}^{(1)}$ of the 1-form symmetry group $\Gamma^{(1)}$.

In this paper, we do *not* study 2-groups that involve action of 0-form symmetry on 1-form symmetry. The 2-groups that we study can be described in terms of a non-closedness condition on the 1-form symmetry background $B_2$ of the form

$$\delta B_2 = B_1^* \Theta \,, \tag{2.1}$$

where $\Theta \in H^3\big(B\mathcal{F}, \Gamma^{(1)}\big)$ is known as the Postnikov class of the 2-group symmetry, $\Gamma^{(1)}$ is the 1-form symmetry group, $\mathcal{F}$ is the 0-form *flavor* symmetry group, $B\mathcal{F}$ is its classifying space and $B_1^*$ is the pullback associated to the map $B_1 : M \to B\mathcal{F}$ from the spacetime manifold $M$ to $B\mathcal{F}$ associated to a background principal bundle for $\mathcal{F}$.

Moreover, in this paper we only study the absence/presence of a particular term of the form $\text{Bock}(w_2)$ in the Postnikov class

$$\Theta = \text{Bock}(w_2) + \cdots \,, \tag{2.2}$$

where $w_2 \in H^2(B\mathcal{F}, \mathcal{Z})$ describes the obstruction class associated to lifting $\mathcal{F}$ bundles to $F$-bundles, where $\mathcal{F} = F/\mathcal{Z}$ and $\mathcal{Z}$ is a subgroup of the center $Z_F$ of $F$ (which therefore is a cover of $\mathcal{F}$). Bock represents the Bockstein homomorphism (which is the connecting homomorphism in the associated long exact sequence in cohomology)

$$\text{Bock}: \qquad H^2(B\mathcal{F}, \mathcal{Z}) \to H^3(B\mathcal{F}, \Gamma^{(1)}) \,, \tag{2.3}$$

associated to a short exact sequence

$$0 \to \Gamma^{(1)} \to \mathcal{E} \to \mathcal{Z} \to 0 \,, \tag{2.4}$$

extending $\mathcal{Z}$ by $\Gamma^{(1)}$. Note that, if the short exact sequence splits, then the Bockstein homomorphism is trivial, we do not obtain a non-trivial contribution of the form $\text{Bock}(w_2)$ to the Postnikov class, and thus there is no 2-group symmetry of this type.

To determine whether a theory has 2-groups of this type it is necessary to compute the 1-form symmetry group $\Gamma^{(1)}$, as well as the flavor symmetry group $\mathcal{F}$, and thus the discrete subgroup $\mathcal{Z}$ such that

$$\mathcal{F} = F/\mathcal{Z} \,, \tag{2.5}$$

with $F$ a cover of the flavor symmetry group. As we describe below, it is possible to do so from the spectrum of the theory and determine these groups as well as the short exact sequence (2.4). Necessary conditions for there to be a 2-group of this type are that the flavor symmetry group $\mathcal{F}$ is not simply-connected, the non-triviality of the 1-form symmetry and that the sequence (2.4) does not split. Another condition is also that the associated Postnikov class is non-vanishing.

## 2.2 Computing 2-Groups From Properties of Line Defects

A more physically intuitive way to deduce the presence of this term in the Postnikov class is by studying line defects and flavor Wilson lines modulo screening [21, 23]:

$$R_1 \longrightarrow \underset{R_2 \otimes R_1^*}{\bullet} R_2 \qquad \Longrightarrow \qquad R_1 \sim R_2$$

Figure 3: A genuine local operator transforming in representation $R_2 \otimes R_1^*$ of the flavor symmetry algebra $\mathfrak{f}$ can be regarded as transforming a flavor Wilson line in representation $R_1$ to a flavor Wilson line in representation $R_2$. The above configuration of the local operator joined to flavor Wilson lines is consistent as it is invariant under gauge transformations of a background flavor connection. If such a local operator exists, then we regard the $R_1$ and $R_2$ flavor Wilson lines to be in the same equivalence class. Such equivalence classes form a group $\widehat{\mathcal{Z}}$ with product operation being tensor product of representations.

$$(L_1, R_1) \longrightarrow \underset{O_{21} \in R_2 \otimes R_1^*}{\bullet} (L_2, R_2) \qquad \Longrightarrow \qquad (L_1, R_1) \sim (L_2, R_2)$$

Figure 4: Now, consider a non-genuine local operator $O_{21}$ transitioning line defect $L_1$ to line defect $L_2$. Say $O_{21}$ transforms as $R_2 \otimes R_1^*$ under the flavor algebra. Then we regard elements $(L_1, R_1)$ and $(L_2, R_2)$ (in the product set of line defects and flavor Wilson lines) to lie in the same equivalence class. Such equivalence classes form a group $\widehat{\mathcal{E}}$ with product operation being OPE and tensor product of representations.

- $\Gamma^{(1)}$ is computed as the Pontryagin dual of the group $\widehat{\Gamma}^{(1)} = \mathrm{Hom}(\Gamma^{(1)}, U(1))$ formed by equivalence classes of line defects modulo screenings due to non-genuine local operators[4] (see figure 2), where one does not include flavor Wilson lines or any information about flavor charges of the local operators.

- The flavor symmetry group $\mathcal{F}$ has the key property that the representations formed by *genuine* local operators are allowed representations for $\mathcal{F}$, but not allowed representations of any other group of the form $F' = \mathcal{F}/Z'$ with $Z'$ being some non-trivial subgroup of the center $Z_{\mathcal{F}}$ of $\mathcal{F}$.

- On the other hand, the group $F$ can be taken to be any covering group of $\mathcal{F}$ such that the representations formed by genuine and *non-genuine* local operators are allowed representations for $F$.

- We can write $\mathcal{F} = F/\mathcal{Z}$, which provides the definition for $\mathcal{Z}$. The group $\mathcal{Z}$ can also be understood as the Pontryagin dual of the group $\widehat{\mathcal{Z}}$ formed by equivalence classes of flavor Wilson lines modulo screenings due to genuine local operators (see figure 3).

- $\mathcal{E}$ is computed as the Pontryagin dual of the group $\widehat{\mathcal{E}}$ formed by equivalence classes of line defects plus flavor Wilson lines modulo screenings due to genuine and non-genuine local operators (see figure 4).

- $\widehat{\mathcal{Z}}$ naturally forms a subgroup of $\widehat{\mathcal{E}}$, leading to the short exact sequence

$$0 \to \widehat{\mathcal{Z}} \to \widehat{\mathcal{E}} \to \widehat{\Gamma}^{(1)} \to 0 \,, \tag{2.6}$$

  whose Pontryagin dual produces the key short exact sequence (2.4).

---

[4] We remind the reader that a non-genuine local operator is one that is constrained to live at a 0-dimensional end or junction of higher-dimensional defects. On the other hand, a genuine local operator exists independently of higher-dimensional defects.

## 2.3  2-Groups for Gauge Theories in $d$ Dimensions

Another alternate way of deducing such 2-groups opens up if the theory under study admits a gauge theory description with a non-abelian gauge algebra $\mathfrak{g}$ and gauge group $G$.

For $d \leq 3$, the gauge theory is UV complete on its own, and we study this UV complete theory. We do not add any Chern-Simons terms (or finite versions thereof) involving either only dynamical fields or dynamical and background fields[5].

For $d = 4$, we study gauge theories having the property that all the gauge couplings have non-positive beta functions. Such a gauge theory is UV complete on its own and we study this UV complete theory. We furthermore assume that the gauge group $G = \mathcal{G}$, where $\mathcal{G}$ is the simply-connected group associated to the gauge algebra $\mathfrak{g}$. This is to ensure that the 2-group does not receive extra "magnetic or dyonic" contributions besides the "electric" contributions coming from matter fields, which further complicate the analysis. See [23] for details on how to handle these extra contributions to the 2-group.

For $d = 5$, we study gauge theories that describe low-energy physics of relevant deformations of 5d $\mathcal{N} = 1$ SCFTs. For $d = 6$, we study gauge theories that describe low-energy physics on the tensor branch of vacua of 6d $\mathcal{N} = (1, 0)$ SCFTs and LSTs.

Under the assumptions discussed above, for $d \leq 4$, we only need to include contributions coming from the matter fields of the gauge theory. However, for $d = 5, 6$ we also need to include further instantonic contributions. Such contributions for 6d theories were discussed in [68] and the contributions relevant for our purposes are described later in this paper (see section 3.2). For 5d theories, see [20, 68] for a detailed description of such contributions.

Pick a $d$-dimensional gauge theory of one of the types discussed above. Let $\psi_i$ be the matter fields in the gauge theory. It transforms in an irrep $R_G^{(i)} \otimes R_F^{(i)}$ of $G \times F$, where $G$ is the gauge group and $F$ is the cover of the flavor group $\mathcal{F}$. Each matter field carries a charge under the center $Z_G$ of the gauge group $G$, and the center $Z_F$ of the cover $F$ of the flavor group $\mathcal{F}$. This is provided by the charge of $R_G^{(i)}$ under $Z_G$ and the charge of $R_F^{(i)}$ under $Z_F$. These charges describe elements $\beta_i$ of $\widehat{Z}_G \times \widehat{Z}_F$, where $\widehat{Z}_G, \widehat{Z}_F$ are Pontryagin duals of $Z_G, Z_F$ respectively.

For $d = 5, 6$, the extra instantonic contributions $a$ similarly provide elements $\beta_a$ of $\widehat{Z}_G \times \widehat{Z}_F$. Similarly, if we drop the restrictions on gauge theories in $d \leq 4$ discussed above, then we obtain extra contributions $a$ which can be incorporated in the same fashion as done above for $d = 5, 6$ gauge theories.

Let $\mathcal{M}$ be the sub-lattice of $\widehat{Z}_G \times \widehat{Z}_F$ generated by $\beta_i$ and $\beta_a$ for all $i, a$. From this we can extract the data relevant for 2-group symmetries as follows:

- $\mathcal{E}$ is the subgroup of $Z_G \times Z_F$ that pairs trivially with $\mathcal{M}$. That is, an element $\alpha \in \mathcal{E}$ iff it acts trivially on all matter fields and extra contributions.

- $\Gamma^{(1)}$ is the subgroup of $\mathcal{E}$ such that $\alpha \in \Gamma^{(1)}$ iff $\pi_F(\alpha) = 0$, where $\pi_F$ is the projection map $\pi_F : Z_G \times Z_F \to Z_F$. In other words, $\Gamma^{(1)} = \mathcal{E} \cap Z_G$.

- $\mathcal{Z}$ is the subgroup of $Z_F$ defined as the image $\pi_F(\mathcal{E})$ of $\mathcal{E}$ under the projection map $\pi_F$. One can easily check that $\mathcal{Z}$ can be identified as $\mathcal{E}/\Gamma^{(1)}$, leading to the key short exact sequence (2.4).

### 2.3.1  Structure Group

The data of $\mathcal{E}$ can be used to assign a *structure group* $\mathcal{S}$ to the gauge theory via

$$\mathcal{S} = \frac{G \times F}{\mathcal{E}} \,. \tag{2.7}$$

---

[5]If these assumptions are violated, then monopole operators produce extra contributions that need to be accounted alongside the matter field contributions. A systematic analysis of these extra contributions in the context of generalized global symmetries will appear elsewhere.

The importance of the structure group is that it describes the full set of gauge and flavor bundles that can be turned on in the gauge theory. A gauge bundle and a flavor bundle can be turned on simultaneously only if they combine to form a bundle for the structure group $\mathcal{S}$.

Let us begin by choosing a background $B_2 \in C^2(M, \Gamma^{(1)})$ which is a 2-cochain valued in $\Gamma^{(1)}$ on spacetime $M$. Also, choose a bundle for the flavor symmetry group $\mathcal{F}$, which comes equipped with a characteristic class $[w_2] \in H^2(M, \mathcal{Z})$ describing the obstruction of lifting the $\mathcal{F}$ bundle to an $F$ bundle. Combine the data of $B_2, w_2$ as follows

$$B_w = i(B_2) + \widetilde{w}_2, \tag{2.8}$$

where $i : C^2(M, \Gamma^{(1)}) \to C^2(M, \mathcal{E})$ induced by $\Gamma^{(1)} \to \mathcal{E}$, and $\widetilde{w}_2 \in C^2(M, \mathcal{E})$ is a 2-cochain lifting $w_2$ from $\mathcal{Z}$ to $\mathcal{E}$. $B_w$ is closed and describes an element $[B_w] \in H^2(M, \mathcal{E})$.

Let us define

$$\Gamma^{(1)'} = \pi_G(\mathcal{E}), \tag{2.9}$$

where

$$\pi_G : \qquad Z_G \times Z_F \to Z_G, \tag{2.10}$$

is the projection map onto $Z_G$. This lets us construct

$$w_2' := \pi_G[B_w] \in H^2(M, \Gamma^{(1)'}), \tag{2.11}$$

which describes the obstruction class of lifting $G/\Gamma^{(1)'}$-gauge bundles to $G$-bundles.

Once a 1-form symmetry background $B_2$ and a flavor background bundle are chosen (which fixes $w_2$), the gauge theory sums over all possible $G/\Gamma^{(1)'}$ bundles with a fixed value of $w_2'$ that is determined in terms of $B_2, w_2$ via (2.11) and (2.8).

One important point to note is that the 1-form symmetry background $B_2$ cannot be chosen independently from the flavor background bundle. This can be seen from the form of $B_w$ appearing in (2.8). Since $B_w$ is closed, applying $\delta$ on both sides of (2.8) leads to the relation[6]

$$\delta B_2 = \text{Bock}(w_2), \tag{2.12}$$

which recovers the fact that 1-form symmetry and flavor symmetry combine to form a 2-group symmetry.

### 2.3.2 Computation Using Charge Matrix

As we have discussed above, there are many key ingredients that go into the determination of 2-group symmetry:

- The (isomorphism classes of) groups $\Gamma^{(1)}$, $\mathcal{E}$ and $\mathcal{Z}$.

- The embedding $i : \Gamma^{(1)} \to \mathcal{E}$, and the projection $\pi : \mathcal{E} \to \mathcal{Z}$.

- The embedding $i_F : \mathcal{Z} \to Z_F$.

The first two ingredients listed above determine the short exact sequence (2.4), which is used for the determination of the precise Bockstein homomorphism to be used in computing the Postnikov class (2.2). On the other hand, the last ingredient listed above determines the flavor symmetry group $\mathcal{F}$ and the obstruction class $w_2$ used in the computation of Postnikov class (2.2).

These ingredients can be computed algorithmically using a charge matrix $\mathcal{M}$, as we discuss in this subsubsection. We can build $\mathcal{M}$ iteratively as follows:

---

[6]See for example [21] for the details on intermediate steps in the calculation.

- Decompose the gauge group center as $Z_G = \prod_{i=1}^{I} \mathbb{Z}_{n_i}$, and the center of the cover of flavor group as $Z_F = \prod_{a=1}^{A} \mathbb{Z}_{n_a}$.

- Start with a diagonal square matrix $\mathcal{M}_I$ of rank $I$. The $i$-th entry on the diagonal of $\mathcal{M}_I$ is $n_i$.

- Take another diagonal matrix $\mathcal{M}_A$ of rank $A$ whose $a$-th diagonal entry is $n_a$ [7]

- Join $\mathcal{M}_I$ and $\mathcal{M}_A$ to make a diagonal matrix $\mathcal{M}_{I+A}$ of rank $I+A$.

- Let $\phi_\alpha$ be different matter fields (and extra non-perturbative contributions). Each $\phi_\alpha$ carries a charge $n_{\alpha,i}$ (mod $n_i$) under $\mathbb{Z}_{n_i}$ and a charge $n_{\alpha,a}$ (mod $n_a$) under $\mathbb{Z}_{n_a}$.

- For each $\alpha$, append a column to $\mathcal{M}_{I+A}$ whose $i$-th entry is $n_{\alpha,i}$ and $a$-th entry is $n_{\alpha,a}$. Let the total number of columns being added be $N$.

After appending all such columns, the resulting matrix of rank $(I+A) \times (I+A+N)$ is what we call the charge matrix $\mathcal{M}$:

$$
\mathcal{M} = \left[ \begin{array}{cc|ccc} \mathcal{M}_I & & n_{1,1} & \cdots & n_{N,1} \\ & & \vdots & & \vdots \\ & & n_{1,I} & \cdots & n_{N,I} \\ \hline & \mathcal{M}_A & \vdots & & \vdots \\ & & n_{1,A} & \cdots & n_{N,A} \end{array} \right] .
\tag{2.13}
$$

We will also need a submatrix $\mathcal{M}_G$ of $\mathcal{M}$,

$$
\mathcal{M} = \left[ \frac{\mathcal{M}_G}{\mathcal{M}_F} \right] ,
\tag{2.14}
$$

which is obtained by keeping only the top $I$ rows of $\mathcal{M}$ and discarding the bottom $A$ rows. The rank of $\mathcal{M}_G$ is $I \times (I+A+N)$.

The **1-form symmetry** $\Gamma^{(1)}$ is obtained by computing Smith Normal Form $\mathrm{SNF}(\mathcal{M}_G)$ of $\mathcal{M}_G$. Each row $i$ of $\mathrm{SNF}(\mathcal{M}_G)$ contains a single non-zero entry $p_i$. Then, we can write

$$
\Gamma^{(1)} = \prod_{i=1}^{I} \mathbb{Z}_{p_i} .
\tag{2.15}
$$

Similarly, the **group** $\mathcal{E}$ is obtained by computing Smith Normal Form $\mathrm{SNF}(\mathcal{M})$ of $\mathcal{M}$. Each row $i$ of $\mathrm{SNF}(\mathcal{M})$ contains a single non-zero entry $q_i$. Then, we can write

$$
\mathcal{E} = \prod_{i=1}^{I+A} \mathbb{Z}_{q_i} .
\tag{2.16}
$$

Now we discuss the computation of the **short exact sequence** (2.4) from the charge matrix $\mathcal{M}$. Firstly, we want to understand the embedding of $\Gamma^{(1)}$ into $\mathcal{E}$. Let us express $\mathrm{SNF}(\mathcal{M}_G)$ in terms of $\mathcal{M}_G$ via two integral square matrices $A_G$ and $B_G$ as follows

$$
\mathrm{SNF}(\mathcal{M}_G) = A_G \cdot \mathcal{M}_G \cdot B_G .
\tag{2.17}
$$

---

[7]Notice that $a$ was used in a different context at the start of this subsection 2.3.

The rank of $A_G$ is $I$ and the rank of $B_G$ is $I+A+N$. $A_G$ encodes the map $\Gamma^{(1)} \to Z_G$, but this map is not of particular relevance to us. We can also implement the transformations performed by $A_G, B_G$ on the full matrix $\mathcal{M}$, which leads to a new matrix $\mathcal{M}'$

$$\mathcal{M}' = \left[ \begin{array}{c|c} A_G & \\ \hline & \mathbb{I}_{A \times A} \end{array} \right] \cdot \mathcal{M} \cdot B_G \,. \tag{2.18}$$

Additionally, define the integral square matrices $A_{\mathcal{E}}, B_{\mathcal{E}}$ via

$$\text{SNF}(\mathcal{M}') = A_{\mathcal{E}} \cdot \mathcal{M}' \cdot B_{\mathcal{E}} \,, \tag{2.19}$$

with the additional constraint that $A_{\mathcal{E}}$ is an upper diagonal matrix with all its diagonal entries being 1. Note that

$$\text{SNF}(\mathcal{M}') = \text{SNF}(\mathcal{M}) \,. \tag{2.20}$$

The rank of $A_{\mathcal{E}}$ is $I+A$ and the rank of $B_{\mathcal{E}}$ is $I+A+N$. The above process captures the embedding $i : \Gamma^{(1)} \to \mathcal{E}$. To see this embedding explicitly we define the matrix $(A_{\mathcal{E}}^{-1})_G$ as the $I \times (I+A)$ rank matrix obtained by deleting the bottom $A$ rows of $A_{\mathcal{E}}^{-1}$:

$$A_{\mathcal{E}}^{-1} = \left[ \begin{array}{c} (A_{\mathcal{E}}^{-1})_G \\ \hline (A_{\mathcal{E}}^{-1})_F \end{array} \right] \,. \tag{2.21}$$

Let us represent $\mathbb{Z}_{p_i}$ as $\mathbb{Z} \pmod{p_i \mathbb{Z}}$. Pick an element $\alpha$ of $\Gamma^{(1)}$. Its projection onto $\mathbb{Z}_{p_i}$ subfactor is described by an integer $\alpha_i \pmod{p_i}$. Let $e_\alpha$ be a rank $I$ row vector whose $i$-th entry is $\alpha_i$. To describe the image $i(\alpha) \in \mathcal{E}$ of $\alpha \in \Gamma^{(1)}$ we compute

$$f_\alpha := e_\alpha R^t \,, \qquad R := \left( P^{-1} (A_{\mathcal{E}}^{-1})_G Q \right)^t \,, \tag{2.22}$$

where $P = \text{diag}(p_i : i = 1, \dots, I)$ and $Q = \text{diag}(q_i : i = 1, \dots, I+A)$, and the superscript $t$ denotes transpose. The projection of $i(\alpha)$ on $\mathbb{Z}_{q_i}$ subfactor of $\mathcal{E}$ is $(f_\alpha)_i \pmod{q_i}$ where $(f_\alpha)_i$ is the $i$-th entry of the rank $I+A$ row vector $f_\alpha$.

We can also compute **the subgroup $\mathcal{Z}$ of the flavor center** in terms of these matrices, using the fact that $\mathcal{Z} = \mathcal{E}/\Gamma^{(1)}$. Now, define a matrix $\mathcal{M}_{\mathcal{Z}}$ by appending the diagonal matrix $Q$ to $R$ as shown below

$$\mathcal{M}_{\mathcal{Z}} = \left[ \begin{array}{c|c} Q & R \end{array} \right] \,. \tag{2.23}$$

This is analogous to the matrix $\mathcal{M}$ we used to compute $\mathcal{E}$ (more precisely its Pontryagin dual $\widehat{\mathcal{E}}$) as a projection from $\widehat{Z}_G \times \widehat{Z}_F$. Here we are computing $\mathcal{Z}$ as a projection from $\mathcal{E}$: computing the Smith Normal Form $\text{SNF}(\mathcal{M}_{\mathcal{Z}})$ of $\mathcal{M}_{\mathcal{Z}}$ directly computes $\mathcal{E}/\Gamma^{(1)}$. Each row $i$ of $\text{SNF}(\mathcal{M}_{\mathcal{Z}})$ contains a single non-zero entry $r_i$. Then, we can write

$$\mathcal{Z} = \prod_{i=1}^{I+A} \mathbb{Z}_{r_i} \,. \tag{2.24}$$

To describe the map $\pi : \mathcal{E} \to \mathcal{Z}$, we define integral square matrices $A_{\mathcal{Z}}, B_{\mathcal{Z}}$ via

$$\text{SNF}(\mathcal{M}_{\mathcal{Z}}) = A_{\mathcal{Z}} \cdot \mathcal{M}_{\mathcal{Z}} \cdot B_{\mathcal{Z}} \,, \tag{2.25}$$

with the additional constraint that $A_{\mathcal{Z}}$ is a *lower* diagonal matrix with all its diagonal entries being 1. The rank of $A_{\mathcal{Z}}$ is $I+A$ and the rank of $B_{\mathcal{Z}}$ is $2I+A$. The matrix $A_{\mathcal{Z}}$ encodes the projection $\pi$ as follows. Pick an element $\beta \in \mathcal{E}$ whose projection onto $\mathbb{Z}_{q_i}$ subfactor is described by an integer $\beta_i \pmod{q_i}$. Associate it to a column vector $c_\beta$ whose $i$-th entry is $\beta_i$. Compute

$$d_\beta := A_{\mathcal{Z}} c_\beta \,. \tag{2.26}$$

Let the $i$-th entry of $d_\beta$ be $(d_\beta)_i$. Then, the projection of $\pi(\beta) \in \mathcal{Z}$ onto its $\mathbb{Z}_{r_i}$ subfactor is given by $(d_\beta)_i \pmod{r_i}$. Thus, we have reconstructed the full short exact sequence (2.4) using the data of charge matrix $\mathcal{M}$.

Finally, in order to find the **flavor symmetry group** $\mathcal{F}$ and the obstruction class $w_2$ appearing in (2.2), we need to understand the map $i_F : \mathcal{Z} \to Z_F$. Pick an element $\gamma \in \mathcal{Z}$ whose projection onto the $\mathbb{Z}_{r_i}$ subfactor is $\gamma_i \pmod{r_i}$. Let $v_\gamma$ be a rank $I + A$ column vector whose $i$-th entry is $\gamma_i$. Then compute

$$u_\gamma := \mathcal{M}_A [(A_{\mathcal{E}}^t)^{-1} Q^{-1} A_{\mathcal{Z}}^{-1}]_F \, v_\gamma \,, \tag{2.27}$$

where $[(A_{\mathcal{E}}^t)^{-1} Q^{-1} A_{\mathcal{Z}}^{-1}]_F$ is obtained from $(A_{\mathcal{E}}^t)^{-1} Q^{-1} A_{\mathcal{Z}}^{-1}$ by deleting its top $I$ rows and keeping the bottom $A$ rows. $u_\gamma$ is then a rank $A$ column vector. The projection of $i_F(\gamma)$ onto its $\mathbb{Z}_{n_a}$ subfactor is given by $u_{\gamma,a} \pmod{n_a}$. This completely specifies the map $i_F : \mathcal{Z} \to Z_F$.

## 2.4 Example: Spin$(4N + 2)$ with Vector Hypers in General Dimension $d$

Consider a $d$-dimensional supersymmetric (with 8 supercharges) Spin$(4N + 2)$ gauge theory with $N_f$ hypermultiplets transforming in vector representation of Spin$(4N + 2)$. The hypers form the fundamental representation of $\mathfrak{f} = \mathfrak{sp}(N_f)$ flavor symmetry algebra, and so we can choose $F = Sp(N_f)$, which is the simply-connected group associated to $\mathfrak{f}$.

For $d \leq 3$, we can allow arbitrary positive values of $N_f$. For $d = 4$, the non-positivity of the beta function implies that we must have $N_f \leq 4N$. For $d = 5$, we have $N_f \leq 4N - 1$ for the gauge theory to arise (at low energies) from a relevant deformation of a 5d SCFT. For $d = 6$, such a gauge theory can arise (at low energies) on the tensor branch of 6d SCFTs, but not on the tensor branch of 6d LSTs, and we must have $N_f = 4N - 6$. This is necessary for the cancellation of 1-loop irreducible quartic gauge anomalies.

Let us now discuss the extra instantonic contributions we need to take into account for $d = 5, 6$. For $d = 5$, it was shown in [20] that the instantonic contribution can be taken to have charge $(0 \pmod 4, N_f \pmod 2)$ under $Z_G \times Z_F = \mathbb{Z}_4 \times \mathbb{Z}_2$. For $d = 6$, the instantons are dynamical strings whose (particle-like) vibration modes give rise to the relevant instantonic contributions we need to take into account. Such instantonic contributions can be taken to have trivial charges under $Z_G \times Z_F$.

Let us first consider that either of the two possibilities holds:

- $d \neq 5$

- Or $d = 5$ and $N_f$ is even.

Then we do not need to worry about any extra instantonic contributions. The matter content transforms in representation $\boldsymbol{F} \otimes \boldsymbol{F}$ of $G \times F = \text{Spin}(4N + 2) \times Sp(N_f)$. We have $Z_G = \mathbb{Z}_4$ and $Z_F = \mathbb{Z}_2$. The generator $\alpha_G$ of $Z_G = \mathbb{Z}_4$ acts as $-1 \in U(1)$ on the matter field, and the generator $\alpha_F$ of $Z_F = \mathbb{Z}_2$ also acts as $-1 \in U(1)$ on the matter field. Thus, the diagonal combination $\alpha_{GF} = (\alpha_G, \alpha_F) \in Z_G \times Z_F$ of the two generators leaves the matter field invariant. This diagonal combination $\alpha_{GF}$ generates the subgroup $\mathcal{E} = \mathbb{Z}_4$ inside $Z_G \times Z_F$. Moreover, we have $\pi_F(2\alpha_{GF}) = 0$, and hence $\Gamma^{(1)} = \mathbb{Z}_2$ is the $\mathbb{Z}_2$ subgroup of $\mathcal{E}$ generated by $2\alpha_{GF}$. Thus, $\mathcal{Z} = \mathcal{E}/\Gamma^{(1)} = \mathbb{Z}_2$, which implies that the flavor symmetry group $\mathcal{F}$ of the theory is

$$\mathcal{F} = F/\mathcal{Z} = Sp(N_f)/\mathbb{Z}_2 \equiv PSp(N_f) \,. \tag{2.28}$$

The key short exact sequence (2.4) becomes

$$0 \to \mathbb{Z}_2 \to \mathbb{Z}_4 \to \mathbb{Z}_2 \to 0 \,, \tag{2.29}$$

which does not split. This leads to a non-trivial 2-group symmetry whose Postnikov class $\Theta$ is

$$\Theta = \text{Bock}(w_2) + \cdots, \tag{2.30}$$

where $w_2 \in H^2(B\mathcal{F}, \mathbb{Z}_2) = H^2(BPSp(N_f), \mathbb{Z}_2)$ is the obstruction class for lifting $\mathcal{F} = PSp(N_f)$ bundles to $F = Sp(N_f)$ bundles, and Bock is the Bockstein homomorphism associated to (2.29). We have $H^3(BPSp(N_f), \mathbb{Z}_2) = \mathbb{Z}_2$ generated by an element $w_3$, and we can identify $\text{Bock}(w_2) = w_3$.

Now consider $d = 5$ and $N_f$ odd. In this case, $\alpha_{GF}$ does not leave the instanton invariant. We instead have $\mathcal{E} = \mathbb{Z}_2$ generated by $(2\alpha_G, 0) \in Z_G \times Z_F$, which can be identified with the 1-form symmetry $\Gamma^{(1)} = \mathbb{Z}_2$. Thus we have $\mathcal{Z} = \mathbb{Z}_1$, implying that the flavor symmetry group is

$$\mathcal{F} = Sp(N_f), \tag{2.31}$$

and there is no 2-group symmetry.

Now let us derive the above results for the first case using the charge matrix. In that case, the charge matrix is

$$\mathcal{M} = \left[\begin{array}{cc|c} 4 & 0 & 2 \\ \hline 0 & 2 & 1 \end{array}\right]. \tag{2.32}$$

Following the algorithm presented above, we can compute

$$\text{SNF}(\mathcal{M}_G) = \begin{pmatrix} 0 & 0 & 2 \end{pmatrix}, \quad \mathcal{M}' = \begin{pmatrix} 0 & 0 & 2 \\ -2 & 2 & 1 \end{pmatrix}, \quad \text{SNF}(\mathcal{M}') = \begin{pmatrix} 4 & 0 & 0 \\ 0 & 0 & 1 \end{pmatrix}, \tag{2.33}$$

from which we can read off $\Gamma^{(1)} = \mathbb{Z}_2$ and $\mathcal{E} = \mathbb{Z}_4 \times \mathbb{Z}_1 = \mathbb{Z}_4$. Thus, we have $p_1 = 1, q_1 = 4, q_2 = 1$. To determine the embedding of the 1-form symmetry into $\mathcal{E}$ we compute[8]

$$A_{\mathcal{E}} = \begin{pmatrix} 1 & -2 \\ 0 & 1 \end{pmatrix}, \tag{2.34}$$

using which we find

$$R^t = \begin{pmatrix} 2 & 1 \end{pmatrix}, \tag{2.35}$$

which implies that the image in $\mathcal{E}$ of the generator of $\Gamma^{(1)} = \mathbb{Z}_2$ has a projection of 2 (mod 4) onto the $\mathbb{Z}_4$ subfactor of $\mathcal{E}$, and a projection of 1 (mod 1) = 0 (mod 1) onto the $\mathbb{Z}_1$ subfactor of $\mathcal{E}$. Thus $\Gamma^{(1)} = \mathbb{Z}_2$ embeds as the $\mathbb{Z}_2$ subgroup of $\mathcal{E} = \mathbb{Z}_4$.

To compute $\mathcal{Z}$, we find that

$$\text{SNF}(\mathcal{M}_{\mathcal{Z}}) = \begin{pmatrix} 0 & 0 & 2 \\ 0 & 1 & 0 \end{pmatrix}, \tag{2.36}$$

implying that $\mathcal{Z} = \mathbb{Z}_2 \times \mathbb{Z}_1 = \mathbb{Z}_2$. That is, $r_1 = 2, r_2 = 1$. To compute the projection $\mathcal{E} \rightarrow \mathcal{Z}$, we first compute

$$A_{\mathcal{Z}} = \begin{pmatrix} 1 & 0 \\ 0 & 1 \end{pmatrix}, \tag{2.37}$$

which means that the generator of $\mathbb{Z}_4$ subfactor of $\mathcal{E}$ is mapped to the generator of the $\mathbb{Z}_2$ subfactor of $\mathcal{Z}$. Let us now compute the embedding of $\mathcal{Z}$ into $Z_F$. For this we need the matrix

$$\mathcal{M}_A[(A_{\mathcal{E}}^t)^{-1} Q^{-1} A_{\mathcal{Z}}^{-1}]_F = \begin{pmatrix} 1 & 2 \end{pmatrix}. \tag{2.38}$$

---

[8]There are various possibilities for $A_{\mathcal{E}}$ depending on the value of the matrix $B_{\mathcal{E}}$. Here, and in what follows, we make one such choice. It should be noted that the mathematica code attached with this paper might produce a different choice.

This implies that the generator of $\mathcal{Z} = \mathbb{Z}_2$ maps to the generator of $Z_F = \mathbb{Z}_2$.

This confirms our results derived above without the use of charge matrix. The use of charge matrix to compute (2.4) might seem a bit overkill in this simple example. However, if one deals with a theory involving large number of gauge and flavor algebras, then the use of charge matrix turns out to be very convenient, especially to perform these with a computer. We have performed the calculation using charge matrix for this simple example to illustrate the various objects involved in such a computation. The ancillary mathematica file provides an implementation of this algorithm.

# 3  2-Group Symmetries in 6d SCFTs and LSTs

Although general arguments [15] show that there cannot be continuous 1-form symmetries and 2-groups in 6d SCFTs, it has been shown in [62, 68], that discrete 1-form symmetries can exist in 6d $\mathcal{N} = (1, 0)$ SCFTs and LSTs.

A 6d $\mathcal{N} = (1, 0)$ SCFT or LST is described (at low energies) along its tensor branch by a 6d $\mathcal{N} = (1, 0)$ non-abelian gauge theory. The gauge theory is coupled to massive string-like excitations. Some of these strings can be recognized as instanton strings of the gauge theory. However, a general 6d SCFT or LST can have strings that are not instantons.

Thus, 2-group symmetries of 6d SCFTs and LSTs can be deduced by applying the gauge-theory-based analysis of section 2.3. Along with matter fields, we have to incorporate contributions from the massive strings discussed above. These are discussed in section 3.2.

## 3.1  Construction of 6d SCFTs and LSTs

A uniform construction of all known 6d SCFTs and LSTs is provided by compactifying F-theory on elliptically fibered Calabi-Yau threefolds. Most of the theories can be constructed by using the unfrozen phase of F-theory, while some outlying theories can only be constructed using the frozen phase of F-theory [78–80]. The classification of the theories lying in the unfrozen phase was performed in [81–83] while the frozen phase theories were classified in [84][9].

The theories resulting from both of these classifications can be described in the same graphical language. The theory is described by a connected graph. A node in the graph takes the following form

$$\overset{\mathfrak{g}}{k},\tag{3.1}$$

where $\mathfrak{g}$ is a simple gauge algebra carried by the node and $k \geq 0$ is known as the value of the node. For 6d SCFTs, the set of allowed nodes can be found in Tables 1 and 2 of [53]. These nodes are also allowed for 6d LSTs, but there are a few more allowed nodes for 6d LSTs that are listed in Table 2 of [83]. The value of all of these nodes is $k = 0$.

The nodes can be joined by single or double edges. For 6d SCFTs, the set of allowed edges can be found in Tables 3–5 of [53]. These edges are also allowed for 6d LSTs, but there are a few more allowed edges for 6d LSTs that are listed in section 7.1.2 of [83].

There are two kinds of flavor symmetries: localized and delocalized. The localized flavor symmetries are associated to a single node in the graph, while delocalized flavor symmetries are associated to multiple nodes. We will see later in section 5.3 that only localized flavor symmetries that are continuous and non-abelian can participate in 2-group symmetries of the type studied in this paper (see section 2.1). A flavor symmetry of this type associated to a node of the form (3.1) is depicted by attaching a flavor node encapsulated between two square

---

[9]See also [85] for a classification based on solving consistency conditions for 6d $\mathcal{N} = (1, 0)$ gauge theories.

brackets as shown below

$$\overset{\mathfrak{g}}{\underset{k}{}} \quad\rule[0.5ex]{2em}{0.4pt}\quad [\mathfrak{f}] \quad, \tag{3.2}$$

where $\mathfrak{f}$ is the non-abelian continuous localized flavor symmetry associated to the node (3.1).

For nodes allowed in both 6d SCFTs and LSTs, such flavor symmetries can be found in Tables 1 and 2 of [69]. The nodes allowed only in 6d LSTs all have a Lagrangian description, so such flavor symmetries are obtained simply by computing the flavor algebras rotating the hypermultiplets.

## 3.2 Strings and Corresponding Charges

Each non-flavor node gives rise to a massive dynamical string on the tensor branch. The various vibration modes of such a (closed) string give rise to massive particles. These particles can provide extra charges under $Z_G \times Z_F$ that are not provided by matter hypermultiplets.

In our considerations regarding 2-group symmetries, such extra charges arise only from strings associated to nodes of the form

$$\overset{\mathfrak{sp}(n)}{1} \quad, \tag{3.3}$$

for $n \geq 0$[10].

Let us now describe the extra charges provided by such a string. Let $i$ be the set of nodes (flavor and non-flavor) neighboring such a node, and let $\mathfrak{g}_i$ be the (flavor or gauge) algebra carried by the $i$-th node. For $n > 0$, we have

$$\bigoplus_i \mathfrak{g}_i \subseteq \mathfrak{so}(4n+16). \tag{3.4}$$

Let $\mathcal{R}$ be the representation of $\bigoplus_i \mathfrak{g}_i$ obtained by reducing the spinor irrep $S$ of $\mathfrak{so}(4n+16)$, and let $\mathcal{R} = \bigoplus_a \mathcal{R}_a$ be the irrep decomposition of $\mathcal{R}$. Then, the extra contributions are completely captured by the representations $\mathcal{R}_a$. For $n = 0$, we instead have

$$\bigoplus_i \mathfrak{g}_i \subseteq \mathfrak{e}_8. \tag{3.5}$$

and the extra contributions are captured by representations $\mathcal{R}_a$ that appear in irrep decomposition of the representation $\mathcal{R}$ obtained by reducing the adjoint representation $\mathfrak{e}_8$ under $\bigoplus_i \mathfrak{g}_i$.

## 3.3 List of 6d SCFTs and LSTs with 2-Group Symmetries

The data discussed above is sufficient to completely classify the 6d SCFTs and LSTs that admit 2-group symmetries. The classification is carried out in detail in the next section. Here we describe the final result of the classification.

We find that there are *no* LSTs that carry the type of 2-group symmetries being discussed in this paper. On the other hand, we find seven classes of 6d SCFTs carrying 2-group symmetries, which are displayed in table 1.

---

[10]For $n = 0$, the gauge algebra $\mathfrak{sp}(0)$ is trivial and the associated string is not an instanton for any gauge group.

Table 1: All types of 6d SCFTs consistent with 2-group symmetry. Out of these, only type 1 arises in the unfrozen phase of F-theory, while the other types arise in the frozen phase of F-theory. The frozen phase theories also admit a type IIA brane construction [86, 87].[11]

| Label | Quiver |
|---|---|
| Type 1 | $\mathfrak{so}(4n_1{+}2)_4 - \mathfrak{sp}(n_2)_1 - \mathfrak{so}(4n_3{+}2)_4 \,{-}\,{-}\, \mathfrak{sp}(n_{2R})_1 - \mathfrak{so}(4n_{2R+1}{+}2)_4$ <br> with flavors $[\mathfrak{sp}(m_1)]$, $[\mathfrak{so}(4m_2)]$, $[\mathfrak{sp}(m_3)]$, $[\mathfrak{so}(4m_{2R})]$, $[\mathfrak{sp}(m_{2R+1})]$ |
| Type 2 | top branch: $\mathfrak{so}(4p)_4 - [\mathfrak{sp}(q)]$ attached to $\mathfrak{sp}(n_{2R})$ <br> bottom: $\mathfrak{so}(4n_1{+}2)_4 - \mathfrak{sp}(n_2)_1 - \mathfrak{so}(4n_3{+}2)_4 \,{-}\,{-}\, \mathfrak{sp}(n_{2R})_1 - \mathfrak{so}(4n_{2R+1}{+}2)_4$ <br> with flavors $[\mathfrak{sp}(m_1)]$, $[\mathfrak{so}(4m_2)]$, $[\mathfrak{sp}(m_3)]$, $[\mathfrak{so}(4m_{2R})]$, $[\mathfrak{sp}(m_{2R+1})]$ |
| Type 3 | top branch: $\mathfrak{so}(4p{+}2)_4 - [\mathfrak{sp}(q)]$ attached to $\mathfrak{sp}(n_{2R})$ <br> bottom: $\mathfrak{so}(4n_1)_4 - \mathfrak{sp}(n_2)_1 - \mathfrak{so}(4n_3)_4 \,{-}\,{-}\, \mathfrak{so}(4n_{2R-1})_4 - \mathfrak{sp}(n_{2R})_1 - \mathfrak{so}(4n_{2R+1}{+}2)_4$ <br> with flavors $[\mathfrak{sp}(m_1)]$, $[\mathfrak{so}(4m_2)]$, $[\mathfrak{sp}(m_3)]$, $[\mathfrak{sp}(m_{2R-1})]$, $[\mathfrak{so}(4m_{2R})]$, $[\mathfrak{sp}(m_{2R+1})]$ |
| Type 3′ | top branch: $\mathfrak{so}(4p{+}2)_4 - [\mathfrak{sp}(q)]$ attached to $\mathfrak{sp}(n_{2R})$ <br> bottom: $\mathfrak{sp}(n_2)_1 - \mathfrak{so}(4n_3)_4 \,{-}\,{-}\, \mathfrak{so}(4n_{2R-1})_4 - \mathfrak{sp}(n_{2R})_1 - \mathfrak{so}(4n_{2R+1}{+}2)_4$ <br> with flavors $[\mathfrak{so}(4m_2)]$, $[\mathfrak{sp}(m_3)]$, $[\mathfrak{sp}(m_{2R-1})]$, $[\mathfrak{so}(4m_{2R})]$, $[\mathfrak{sp}(m_{2R+1})]$ |
| Type 4 | top branch: $[\mathfrak{sp}(q_1)] - \mathfrak{so}(4p_1)_4 - \mathfrak{sp}(p_2)_1 - [\mathfrak{so}(4q_2)]$ attached to $\mathfrak{so}(4n_3{+}2)$ <br> bottom: $\mathfrak{so}(4n_1{+}2)_4 - \mathfrak{sp}(n_2)_1 - \mathfrak{so}(4n_3{+}2)_4 - \mathfrak{sp}(n_4)_1 - \mathfrak{so}(4n_5{+}2)_4$ <br> with flavors $[\mathfrak{sp}(m_1)]$, $[\mathfrak{so}(4m_2)]$, $[\mathfrak{sp}(m_3)]$, $[\mathfrak{so}(4m_4)]$, $[\mathfrak{sp}(m_5)]$ |
| Type 5 | $\mathfrak{su}(2n_1)_2 - \mathfrak{su}(2n_2)_2 \,{-}\,{-}\, \mathfrak{su}(2n_R)_2 - \mathfrak{so}(4n{+}2)_4$ <br> with flavors $[\mathfrak{su}(2m_1)]$, $[\mathfrak{su}(2m_2)]$, $[\mathfrak{su}(2m_R)]$, $[\mathfrak{sp}(2m)]$ |
| Type 6 | $\mathfrak{su}(2p)_2 - \mathfrak{so}(4n_1{+}2)_4 - \mathfrak{sp}(n_2)_1 \,{-}\,{-}\, \mathfrak{sp}(n_{2R})_1 - \mathfrak{so}(4n_{2R+1}{+}2)_4$ <br> with flavors $[\mathfrak{su}(2q)]$, $[\mathfrak{sp}(m_1)]$, $[\mathfrak{so}(4m_2)]$, $[\mathfrak{so}(4m_{2R})]$, $[\mathfrak{sp}(m_{2R+1})]$ |

---

[11]We do not believe that there is any particular physical significance to the fact that the majority of our examples arise in the frozen phase of F-theory, instead suggest this could be a geometric property of the F-theory

### 3.3.1 Classification of Allowed Ranks

We now supply the information in table 1 with some more details on the allowed ranks of the algebras involved. The allowed gauge groups will be discussed in the next subsection.

**Type 1**

$$\mathfrak{so}(4n_1+2) \underset{4}{\quad} \mathfrak{sp}(n_2) \underset{1}{\quad} \mathfrak{so}(4n_3+2) \underset{4}{\quad -----\quad} \mathfrak{sp}(n_{2R}) \underset{1}{\quad} \mathfrak{so}(4n_{2R+1}+2) \underset{4}{\quad}$$

$$[\mathfrak{sp}(m_1)] \qquad [\mathfrak{so}(4m_2)] \qquad [\mathfrak{sp}(m_3)] \qquad [\mathfrak{so}(4m_{2R})] \qquad [\mathfrak{sp}(m_{2R+1})] \tag{3.6}$$

Fixing $n_1$ and $\{m_i, i \neq 2R+1\}$ fixes all other nodes:

$$n_2 = 4n_1 - 6 - m_1,$$
$$4n_{2i+1} + 2 = 4n_{2i} + 16 - 4m_{2i} - (4n_{2i-1} + 2), (i = 1, \ldots, R). \tag{3.7}$$
$$n_{2i} = 4n_{2i-1} - m_{2i-1} - 6 - n_{2i-2}, (i = 2, \ldots, R),$$

and

$$m_{2R+1} = 4n_{2R+1} - 6 - n_{2R}. \tag{3.8}$$

**Type 2**

$$\mathfrak{so}(4p) \underset{4}{\quad} [\mathfrak{sp}(q)]$$

$$\mathfrak{so}(4n_1+2) \underset{4}{\quad} \mathfrak{sp}(n_2) \underset{1}{\quad} \mathfrak{so}(4n_3+2) \underset{4}{\quad ----- \quad} \mathfrak{sp}(n_{2R}) \underset{1}{\quad} \mathfrak{so}(4n_{2R+1}+2) \underset{4}{\quad}$$

$$[\mathfrak{sp}(m_1)] \qquad [\mathfrak{so}(4m_2)] \qquad [\mathfrak{sp}(m_3)] \qquad [\mathfrak{so}(4m_{2R})] \qquad [\mathfrak{sp}(m_{2R+1})] \tag{3.9}$$

Fixing $n_1$ and all $m_i$ fixes the quiver entirely:

$$n_2 = 4n_1 - 6 - m_1,$$
$$4n_{2i-1} + 2 = 4n_{2i-2} + 16 - 4m_{2i-2} - (4n_{2i-3} + 2), i = 2, \ldots, R$$
$$n_{2i} = 4n_{2i-1} - 6 - m_{2i-1} - n_{2i-2}, i = 2, \ldots, R$$
$$4n_{2R+1} - 6 = n_{2R} + m_{2R+1},$$
$$4p = 4n_{2R} + 16 - 4m_{2R} - (4n_{2R-1} + 2) - (4n_{2R+1} + 2),$$
$$q = 4p - 8 - n_{2R}. \tag{3.10}$$

**Type 3**

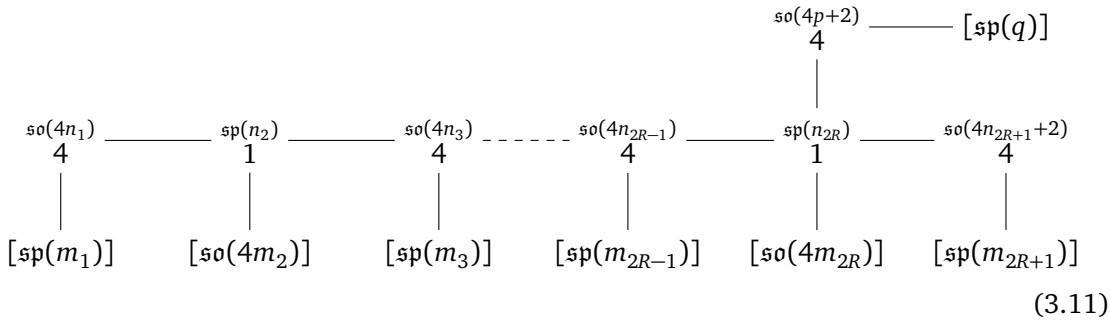

$$\tag{3.11}$$

---

compactifications themselves.

Fixing $n_1$ and all $m_i$ fixes all nodes of these quivers:

$$
\begin{aligned}
n_2 &= 4n_1 - 8 - m_1 \,, \\
4n_{2i-1} &= 4n_{2i-2} + 16 - 4m_{2i-2} - 4n_{2i-3} \,, \quad i = 2,\dots,R. \\
n_{2i} &= 4n_{2i-1} - 8 - m_{2i-1} - n_{2i-2} \,, \quad i = 2,\dots,R. \,, \\
4n_{2R+1} - 6 &= n_{2R} + m_{2R+1} \,, \\
4p + 2 &= 4n_{2R} + 16 - 4m_{2R} - (4n_{2R-1} + 2) - (4n_{2R+1} + 2) \,, \\
q &= 4p - 6 - n_{2R} \,.
\end{aligned}
\tag{3.12}
$$

**Type 3'** Note that the quivers of type $3'$ can be formed from those of type 3 by deleting the left-most $\mathfrak{so}$ node and its associated flavor $\mathfrak{sp}$ node. The ranks can be fixed exactly as above for type 3, except now written in terms of $n_2$ and all $m_i$.

**Type 4**

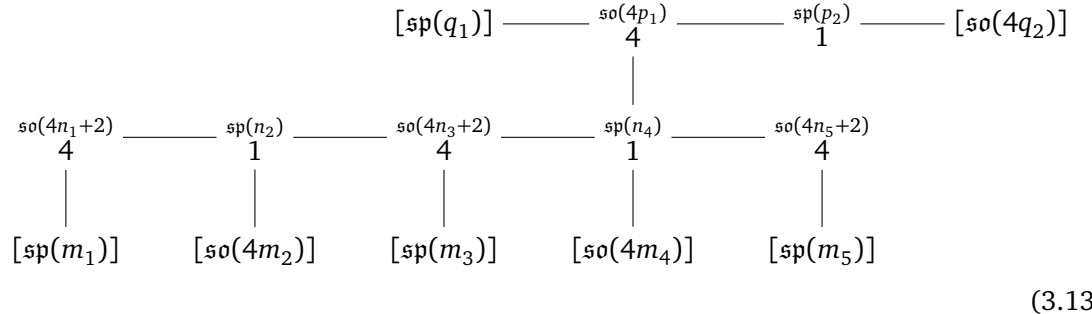

$$
\tag{3.13}
$$

By fixing $n_1, q_1$ and all $m_i$, we fix all other ranks:

$$
\begin{aligned}
n_2 &= 4n_1 - 6 - m_1 \,, \\
4n_3 + 2 &= 4n_2 + 16 - 4m_2 - (4n_1 + 2) \,, \\
n_4 &= 4n_3 - 6 - m_3 - n_2 \,, \\
4n_5 - 6 &= n_4 + m_5 \,, \\
4p_1 &= 4n_4 + 16 - (4n_3 + 2) - (4n_5 + 2) - 4m_4 \,, \\
p_2 &= 4p_1 - 8 - q_1 - n_4 \,, \\
4q_2 &= 4p_2 + 16 - 4p_1 \,.
\end{aligned}
\tag{3.14}
$$

**Type 5**

$$
\begin{array}{ccccccc}
\underset{2}{\mathfrak{su}(2n_1)} & \text{---} & \underset{2}{\mathfrak{su}(2n_2)} & \text{-----} & \underset{2}{\mathfrak{su}(2n_R)} & \text{---} & \underset{4}{\mathfrak{so}(4n+2)} \\
\mid & & \mid & & \mid & & \mid \\
[\mathfrak{su}(2m_1)] & & [\mathfrak{su}(2m_2)] & & [\mathfrak{su}(2m_R)] & & [\mathfrak{sp}(2m)]
\end{array}
\tag{3.15}
$$

We can fix all ranks by fixing $n_1$ and $\{m_i\}$:

$$
\begin{aligned}
2n_2 &= 4n_1 - 2m_1 \,, \\
2n_i &= 4n_{i-1} - 2m_{i-1} - 2n_{i-2} \,, \quad i \neq 1 \,, \\
4n + 2 &= 4n_R - 2n_{R-1} - 2m_R \,, \\
2m &= 4n - 6 - 2n_R \,.
\end{aligned}
\tag{3.16}
$$

**Type 6**

$$
\begin{array}{ccccccccc}
\overset{\mathfrak{su}(2p)}{2} & \rule[0.5ex]{2em}{0.4pt} & \overset{\mathfrak{so}(4n_1+2)}{4} & \rule[0.5ex]{2em}{0.4pt} & \overset{\mathfrak{sp}(n_2)}{1} & \dashrule & \overset{\mathfrak{sp}(n_{2R})}{1} & \rule[0.5ex]{2em}{0.4pt} & \overset{\mathfrak{so}(4n_{2R+1}+2)}{4} \\
| & & | & & | & & | & & | \\
[\mathfrak{su}(2q)] & & [\mathfrak{sp}(m_1)] & & [\mathfrak{so}(4m_2)] & & [\mathfrak{so}(4m_{2R})] & & [\mathfrak{sp}(m_{2R+1})]
\end{array}
\tag{3.17}
$$

In this type, all ranks can be fixed by fixing $p, q$ and $\{m_i, i \neq 2R+1\}$:

$$
\begin{aligned}
4n_1 + 2 &= 4p - 2q\,, \\
n_2 &= 4n_1 - 6 - m_1 - 2p\,, \\
4n_{2i-1} + 2 &= 4n_{2i-2} + 16 - 4m_{2i-2} - (4n_{2i-3} + 2)\,, \quad i = 2,,\ldots,R+1\,. \\
n_{2i} &= 4n_{2i-1} - 6 - m_{2i-1} - n_{2i-2}\,, \quad i = 2,\ldots,R\,. \\
m_{2R+1} &= 4n_{2R+1} - 6 - n_{2R}\,.
\end{aligned}
\tag{3.18}
$$

### 3.3.2 Classification of Allowed Gauge Groups

In this subsection, we classify, for each of the above 7 types, the allowed choices of gauge groups that are consistent with the existence of 2-group symmetry. As will be discussed in section 5, there is always at least one choice of gauge group, which is

$$
G = \prod_i G_i\,,
\tag{3.19}
$$

where $i$ parametrizes various non-flavor nodes and $G_i$ denotes the simply-connected group associated to the gauge algebra $\mathfrak{g}_i$ of the node $i$.

To determine other allowed choices of gauge groups, we need to first determine the 1-form symmetry $\Gamma^{(1)}$ for theories obtained by equipping all 7 types with the above choice (3.19) of gauge group. We find that:

$$
\Gamma^{(1)} = \begin{cases} \mathbb{Z}_2\,, & \text{Types } 1, 3', 4, 5, 6\,, \\ \mathbb{Z}_2 \times \mathbb{Z}_2\,, & \text{Types } 2, 3\,. \end{cases}
\tag{3.20}
$$

Other choices of gauge groups are obtained by gauging subgroups of $\Gamma^{(1)}$. For types 1, $3'$, 4, 5 and 6, the 1-form symmetry is $\mathbb{Z}_2$, which participates in the 2-group symmetry. Gauging this 1-form symmetry removes the 2-group symmetry. Thus, for types 1, $3'$, 4, 5 and 6, there are no allowed choice of gauge groups other than (3.19) that gives rise to a theory with 2-group symmetry.

For types 2 and 3, the 1-form symmetry is $\mathbb{Z}_2^{(2g)} \times \mathbb{Z}_2^{(n2g)}$. The $\mathbb{Z}_2^{(2g)}$ factor participates in 2-group, while the $\mathbb{Z}_2^{(n2g)}$ *does not*. Thus, we can gauge $\mathbb{Z}_2^{(n2g)}$ subgroup of $\Gamma^{(1)}$ or the diagonal $\mathbb{Z}_2$ inside $\Gamma^{(1)} = \mathbb{Z}_2^{(2g)} \times \mathbb{Z}_2^{(n2g)}$ without destroying 2-group symmetry. Hence, there are two more allowed choices of gauge groups other than the choice (3.19).

For type 2, the first additional choice of gauge group is

$$
\prod_{i=1}^{R} \mathrm{Spin}(4n_{2i-1} + 2) \times \prod_{i=1}^{R} Sp(n_{2i}) \times \frac{\mathrm{Spin}(4n_{2R+1} + 2) \times \mathrm{Spin}(4p)}{\mathbb{Z}_2}\,,
\tag{3.21}
$$

where the $\mathbb{Z}_2$ in the denominator can be described as follows: projecting the $\mathbb{Z}_2$ onto center of $\mathrm{Spin}(4n_{2R+1} + 2)$ gives rise to the order 2 element inside its $\mathbb{Z}_4$ center, and projecting the

$\mathbb{Z}_2$ onto center of Spin($4p$) gives rise to the order 2 element inside its $\mathbb{Z}_2^2$ center that does not act on the vector representation of Spin($4p$). The second additional choice of gauge group is

$$\frac{\prod_{i=1}^{R} \text{Spin}(4n_{2i-1} + 2) \times \text{Spin}(4p)}{\mathbb{Z}_2} \times \prod_{i=1}^{R} Sp(n_{2i}) \times \text{Spin}(4n_{2R+1} + 2), \tag{3.22}$$

where the $\mathbb{Z}_2$ in the denominator can be described as follows: projecting the $\mathbb{Z}_2$ onto center of Spin($4n_{2i-1} + 2$) for any $i$ gives rise to the order 2 element inside its $\mathbb{Z}_4$ center, and projecting the $\mathbb{Z}_2$ onto center of Spin($4p$) gives rise to the order 2 element inside its $\mathbb{Z}_2^2$ center that does not act on the vector representation of Spin($4p$).

For type 3, the first additional choice of gauge group is

$$\frac{\prod_{i=1}^{R} \text{Spin}(4n_{2i-1}) \times \text{Spin}(4p + 2)}{\mathbb{Z}_2} \times \prod_{i=1}^{R} Sp(n_{2i}) \times \text{Spin}(4n_{2R+1} + 2), \tag{3.23}$$

where the $\mathbb{Z}_2$ in the denominator can be described as follows: projecting the $\mathbb{Z}_2$ onto center of Spin($4n_{2i-1}$) for any $i$ gives rise to the order 2 element inside its $\mathbb{Z}_2^2$ center that does not act on the vector representation of Spin($4n_{2i-1}$), and projecting the $\mathbb{Z}_2$ onto center of Spin($4p + 2$) gives rise to the order 2 element inside its $\mathbb{Z}_4$ center. The second additional choice of gauge group is

$$\frac{\prod_{i=1}^{R} \text{Spin}(4n_{2i-1}) \times \text{Spin}(4n_{2R+1} + 2)}{\mathbb{Z}_2} \times \prod_{i=1}^{R} Sp(n_{2i}) \times \text{Spin}(4p + 2), \tag{3.24}$$

where the $\mathbb{Z}_2$ in the denominator can be described as follows: projecting the $\mathbb{Z}_2$ onto center of Spin($4n_{2i-1}$) for any $i$ gives rise to the order 2 element inside its $\mathbb{Z}_2^2$ center that does not act on the vector representation of Spin($4n_{2i-1}$), and projecting the $\mathbb{Z}_2$ onto center of Spin($4n_{2R+1}+2$) gives rise to the order 2 element inside its $\mathbb{Z}_4$ center.

### 3.3.3 Flavor Symmetry Groups and Postnikov Classes

A crucial ingredient in the analysis of the 2-groups is the global form of the flavor symmetry group[12]. We now determine the flavor groups for the above types. For types 1–4 (including $3'$) and any choice of gauge groups, the flavor symmetry groups are respectively

$$\begin{aligned}
\mathcal{F}^{\text{Type 1}} &= \frac{\prod_{i=1}^{R} SO(4m_{2i}) \times \prod_{i=1}^{R+1} Sp(m_{2i-1})}{\mathbb{Z}_2}, \\
\mathcal{F}^{\text{Type 2}} &= \frac{\prod_{i=1}^{R} SO(4m_{2i}) \times \prod_{i=1}^{R+1} Sp(m_{2i-1}) \times Sp(q)}{\mathbb{Z}_2}, \\
\mathcal{F}^{\text{Type 3}} &= \frac{\prod_{i=1}^{R} SO(4m_{2i}) \times \prod_{i=1}^{R+1} Sp(m_{2i-1}) \times Sp(q)}{\mathbb{Z}_2}, \\
\mathcal{F}^{\text{Type 3'}} &= \frac{\prod_{i=1}^{R} SO(4m_{2i}) \times \prod_{i=1}^{R+1} Sp(m_{2i+1}) \times Sp(q)}{\mathbb{Z}_2}, \\
\mathcal{F}^{\text{Type 4}} &= \frac{\prod_{i=1}^{2} SO(4m_{2i}) \times \prod_{i=1}^{3} Sp(m_{2i-1}) \times Sp(q_1) \times SO(4q_2)}{\mathbb{Z}_2},
\end{aligned} \tag{3.25}$$

where each subfactor in numerator has a $\mathbb{Z}_2$ center, and the $\mathbb{Z}_2$ appearing in the denominator is the combined diagonal of all of these $\mathbb{Z}_2$s. Let us define the num(erator) part

$$\mathcal{F}^{\text{Type } i} = \frac{\mathcal{F}_{\text{num}}^{\text{Type } i}}{\mathbb{Z}_2}. \tag{3.26}$$

---

[12]Some aspects of the global form of gauge and flavor have been discussed in F-theory in [88].

The Postnikov class for the 2-group symmetry is then

$$\Theta = \text{Bock}(w_2) + \cdots, \tag{3.27}$$

where $w_2$ is the obstruction class for lifting $\mathcal{F}^{\text{Type } i}$ bundles to $\mathcal{F}^{\text{Type } i}_{\text{num}}$ bundles, and Bock is the Bockstein homomorphism associated to the short exact sequence

$$0 \rightarrow \mathbb{Z}_2 \rightarrow \mathbb{Z}_4 \rightarrow \mathbb{Z}_2 \rightarrow 0. \tag{3.28}$$

For types 5 and 6, we do not determine the full flavor symmetry group. However, we can still describe the Postnikov class, which can be written again as in (3.27) with Bockstein homomorphism associated to (3.28). The obstruction class $w_2$ can be identified for type 5 as the obstruction for lifting

$$\mathcal{F}^{\text{Type } 5}_{\text{relevant}} = \frac{\prod_{i=1}^{R} SU(2m_i) \times Sp(2m)}{\mathbb{Z}_2}, \tag{3.29}$$

bundles to

$$\mathcal{F}^{\text{Type } 5}_{\text{relevant, num}} = \prod_{i=1}^{R} SU(2m_i) \times Sp(2m), \tag{3.30}$$

bundles. For type 6, $w_2$ can be identified as the obstruction for lifting

$$\mathcal{F}^{\text{Type } 6}_{\text{relevant}} = \frac{\prod_{i=1}^{R} SO(4m_{2i}) \times \prod_{i=1}^{R+1} Sp(m_{2i-1}) \times SU(2q)}{\mathbb{Z}_2}, \tag{3.31}$$

bundles to

$$\mathcal{F}^{\text{Type } 6}_{\text{relevant, num}} = \prod_{i=1}^{R} SO(4m_{2i}) \times \prod_{i=1}^{R+1} Sp(m_{2i-1}) \times SU(2q), \tag{3.32}$$

bundles. We can write the flavor symmetry group for types 5 and 6 as

$$\mathcal{F}^{\text{Type } i} = \frac{\mathcal{F}^{\text{Type } i}_{\text{relevant}} \times \Gamma}{Z}, \tag{3.33}$$

where $\Gamma$ is an abelian group (involving both continuous and finite factors), and $Z$ is a subgroup of $Z^{\text{Type } i}_{\text{relevant}} \times \Gamma$ where $Z^{\text{Type } i}_{\text{relevant}}$ is the center of $\mathcal{F}^{\text{Type } i}_{\text{relevant}}$. The obstruction class $w_2$ for types 5 and 6 can also be recognized as the obstruction for lifting $\mathcal{F}^{\text{Type } i}$ bundles to

$$\mathcal{F}^{\text{Type } i}_{\text{num}} = \frac{\mathcal{F}^{\text{Type } i}_{\text{relevant, num}} \times \Gamma}{Z}, \tag{3.34}$$

bundles.

### 3.4 Mixed 0-Form 3-Form Anomaly Dual to 2-Group Symmetry

In general $d$ dimensions, gauging a 1-form symmetry participating in a 2-group symmetry, leads to a dual $(d-3)$-form symmetry which instead has a mixed 't Hooft anomaly with the 0-form symmetry group [2].

For the above discussed theories in 6d, we have a $\mathbb{Z}_2$ 1-form symmetry participating in 2-group symmetry. Gauging this $\mathbb{Z}_2$ 1-form symmetry results in a mixed anomaly between $\mathcal{F}^{\text{Type } i}$ 0-form flavor symmetry and the dual $\mathbb{Z}_2$ 3-form symmetry. The associated anomaly theory is

$$I_7 = \int B_4 \cup \text{Bock}(w_2), \tag{3.35}$$

where $w_2$ is the obstruction class appearing in the Postnikov class (3.27), and $B_4$ is the background field for the 3-form symmetry.

The 6d theories having such a mixed anomaly have the following gauge groups:

- For type 1, we have the following gauge group:

$$\frac{\prod_{i=1}^{R+1}\mathrm{Spin}(4n_{2i-1}+2)}{\mathbb{Z}_2}\times\prod_{i=1}^{R}Sp(n_{2i}),\qquad(3.36)$$

where $\mathbb{Z}_2$ is the combined diagonal of the $\mathbb{Z}_2$ subgroups of the $\mathbb{Z}_4$ centers of $\mathrm{Spin}(4n_{2i-1}+2)$ groups.

- For type 2, we have two possibilities for gauge groups. The first possibility is

$$\frac{\prod_{i=1}^{R+1}\mathrm{Spin}(4n_{2i-1}+2)}{\mathbb{Z}_2}\times\prod_{i=1}^{R}Sp(n_{2i})\times\mathrm{Spin}(4p),\qquad(3.37)$$

where $\mathbb{Z}_2$ is the combined diagonal of the $\mathbb{Z}_2$ subgroups of the $\mathbb{Z}_4$ centers of $\mathrm{Spin}(4n_{2i-1}+2)$ groups. The second possibility is

$$\frac{\prod_{i=1}^{R+1}\mathrm{Spin}(4n_{2i-1}+2)\times\mathrm{Spin}(4p)}{\mathbb{Z}_2^{(1)}\times\mathbb{Z}_2^{(2)}}\times\prod_{i=1}^{R}Sp(n_{2i}),\qquad(3.38)$$

where $\mathbb{Z}_2^{(1)}$ is the combined diagonal of the $\mathbb{Z}_2$ subgroups of the $\mathbb{Z}_4$ centers of $\mathrm{Spin}(4n_{2i-1}+2)$ groups. $\mathbb{Z}_2^{(2)}$ projects to the $\mathbb{Z}_2$ inside $\mathbb{Z}_2^2$ center of $\mathrm{Spin}(4p)$ that does not act on the vector rep, and the $\mathbb{Z}_2$ subgroup of the $\mathbb{Z}_4$ center of $\mathrm{Spin}(4n_{2R+1}+2)$.

- For type 3, we have two possibilities for gauge groups. The first possibility is

$$\prod_{i=1}^{R}\mathrm{Spin}(4n_{2i-1})\times\prod_{i=1}^{R}Sp(n_{2i})\times\frac{\mathrm{Spin}(4p+2)\times\mathrm{Spin}(4n_{2R+1}+2)}{\mathbb{Z}_2},\qquad(3.39)$$

where $\mathbb{Z}_2$ is the combined diagonal of the $\mathbb{Z}_2$ subgroups of the $\mathbb{Z}_4$ centers of $\mathrm{Spin}(4n_{2R+1}+2)$ and $\mathrm{Spin}(4p+2)$ groups. The second possibility is

$$\frac{\prod_{i=1}^{R}\mathrm{Spin}(4n_{2i-1})\times\mathrm{Spin}(4n_{2R+1}+2)\times\mathrm{Spin}(4p+2)}{\mathbb{Z}_2^{(1)}\times\mathbb{Z}_2^{(2)}}\times\prod_{i=1}^{R}Sp(n_{2i}),\qquad(3.40)$$

where $\mathbb{Z}_2^{(1)}$ is the combined diagonal of the $\mathbb{Z}_2$ subgroups of the $\mathbb{Z}_4$ centers of $\mathrm{Spin}(4n_{2R+1}+2)$ and $\mathrm{Spin}(4p+2)$ groups. $\mathbb{Z}_2^{(2)}$ projects to the $\mathbb{Z}_2$ inside $\mathbb{Z}_2^2$ center of $\mathrm{Spin}(4n_{2i-1})$ that does not act on its vector rep, and the $\mathbb{Z}_2$ subgroup of the $\mathbb{Z}_4$ center of $\mathrm{Spin}(4n_{2R+1}+2)$.

- For type $3'$, we have the following gauge group:

$$\prod_{i=1}^{R}\mathrm{Spin}(4n_{2i+1})\times\prod_{i=1}^{R}Sp(n_{2i})\times\frac{\mathrm{Spin}(4p+2)\times\mathrm{Spin}(4n_{2R+1}+2)}{\mathbb{Z}_2},\qquad(3.41)$$

where $\mathbb{Z}_2$ in the denominator is the diagonal $\mathbb{Z}_2$ of the $\mathbb{Z}_2$ centers of $\mathrm{Spin}(4p+2)$ and $\mathrm{Spin}(4n_{2R+1}+2)$).

- For type 4, we have the following gauge group:

$$\frac{\prod_{i=1}^{3}\mathrm{Spin}(4n_{2i-1}+2)}{\mathbb{Z}_2}\times\prod_{i=1}^{2}Sp(n_{2i})\times\mathrm{Spin}(4p_1)\times Sp(p_2),\qquad(3.42)$$

where $\mathbb{Z}_2$ is the combined diagonal of the $\mathbb{Z}_2$ subgroups of the $\mathbb{Z}_4$ centers of $\mathrm{Spin}(4n_{2i-1}+2)$ groups.

- For type 5, we have the following gauge group:

$$\prod_{i=1}^{R} SU(2n_{2i}) \times SO(4n+2).$$ (3.43)

- For type 6, we have the following gauge group:

$$\frac{\prod_{i=1}^{R+1} \text{Spin}(4n_{2i-1}+2)}{\mathbb{Z}_2} \times \prod_{i=1}^{R} Sp(n_{2i}) \times SU(2p),$$ (3.44)

where $\mathbb{Z}_2$ is the combined diagonal of the $\mathbb{Z}_2$ subgroups of the $\mathbb{Z}_4$ centers of Spin$(4n_{2i-1}+2)$ groups.

### 3.5 A Quiver Example

In this subsection, we discuss in some detail the calculation of 2-group symmetry in the simplest quiver example among the seven types of theories appearing above. Consider the 6d theory:

$$
\begin{array}{ccc}
\underset{4}{\mathfrak{so}(4n_1+2)} \underline{\hspace{1.2cm}} \underset{1}{\mathfrak{sp}(n_2)} \underline{\hspace{1.2cm}} \underset{4}{\mathfrak{so}(4n_3+2)} \\
| \qquad\qquad | \qquad\qquad | \\
[\mathfrak{sp}(m_1)] \qquad [\mathfrak{so}(4m_2)] \qquad [\mathfrak{sp}(m_3)]
\end{array}
,$$ (3.45)

with the gauge group chosen to be

$$G = \text{Spin}(4n_1+2) \times Sp(n_2) \times \text{Spin}(4n_3+2).$$ (3.46)

Its center is

$$Z_G = Z_1 \times Z_2 \times Z_3 = \mathbb{Z}_4 \times \mathbb{Z}_2 \times \mathbb{Z}_4.$$ (3.47)

The subgroup of $Z_G$ that leaves the hypermultiplets invariant is

$$\widetilde{\Gamma}^{(1)} = \mathbb{Z}_2 \times \mathbb{Z}_2,$$ (3.48)

where the first $\mathbb{Z}_2$ factor is the $\mathbb{Z}_2$ subgroup of $Z_1 = \mathbb{Z}_4$, while the second $\mathbb{Z}_2$ factor is the $\mathbb{Z}_2$ subgroup of $Z_3 = \mathbb{Z}_4$.

However the 1-form symmetry $\Gamma^{(1)}$ is not given by $\widetilde{\Gamma}^{(1)}$. Only the diagonal of the two $\mathbb{Z}_2$ factors in $\widetilde{\Gamma}^{(1)}$ survives as 1-form symmetry of the full theory. This is because the instanton string associated to the $Sp(n_2)$ provides excitations that are charged as bi-spinor of Spin$(4n_1+2) \times$ Spin$(4n_3+2)$. This is only left invariant by the diagonal $\mathbb{Z}_2$ inside $\widetilde{\Gamma}^{(1)}$. Thus, the 1-form symmetry group for this 6d theory is

$$\Gamma^{(1)} = \mathbb{Z}_2.$$ (3.49)

Now, let us compute the 0-form flavor symmetry group $\mathcal{F}$ of this 6d theory. We need to first pick a global form $F$ of the flavor algebra

$$\mathfrak{f} = \mathfrak{sp}(m_1) \oplus \mathfrak{so}(4m_2) \oplus \mathfrak{sp}(m_3),$$ (3.50)

such that all the representations under $\mathfrak{f}$ formed by hypermultiplets and string states are allowed representations of $F$. Let us assume $m_1$, $m_2$ and $m_3$ are all non-zero. Then, looking at the matter content, we find that $F$ must allow for fundamental representations of $\mathfrak{sp}(m_1)$ and $\mathfrak{sp}(m_3)$, and vector representation of $\mathfrak{so}(4m_2)$. The string states are charged as spinor (**S**) and

co-spinor ($C$) irreps of $\mathfrak{so}(4m_2)$, so these irreps should also be allowed by $F$. Thus, we must pick

$$F = Sp(m_1) \times \text{Spin}(4m_2) \times Sp(m_3),\tag{3.51}$$

whose center is

$$Z_F = \mathbb{Z}_2 \times \mathbb{Z}_2^2 \times \mathbb{Z}_2.\tag{3.52}$$

In order to now compute $\mathcal{E}$, we need all the charges contributed by the $Sp(n_2)$ instanton string. This string provides extra states transforming in representation

$$\boldsymbol{SSS} \oplus \boldsymbol{SCC} \oplus \boldsymbol{CSC} \oplus \boldsymbol{CCS},\tag{3.53}$$

of $\text{Spin}(4n_1+2) \times \text{Spin}(4n_3+2) \times \text{Spin}(4m_2)$. These states and hypermultiplets are left invariant by

$$\mathcal{E} = \mathbb{Z}_4 \times \mathbb{Z}_2,\tag{3.54}$$

subgroup of $Z_G \times Z_F$. The projection of the $\mathbb{Z}_4$ factor in $\mathcal{E}$ on the centers of $Sp(m_1)$, $Sp(n_2)$ and $Sp(m_3)$ is $\mathbb{Z}_2$, on the centers of $\text{Spin}(4n_1+2)$ and $\text{Spin}(4n_3+2)$ is $\mathbb{Z}_4$, and the center of $\text{Spin}(4m_2)$ is the $\mathbb{Z}_2$ that acts on spinor irrep but does not act on cospinor irrep. The projection of the $\mathbb{Z}_2$ factor in $\mathcal{E}$ on the centers of $Sp(m_1)$, $\text{Spin}(4n_1+2)$, $Sp(n_2)$ and $Sp(m_3)$ is trivial $\mathbb{Z}_1$, on the center of $\text{Spin}(4n_3+2)$ is $\mathbb{Z}_2$, and the center of $\text{Spin}(4m_2)$ is the $\mathbb{Z}_2$ that does not act on the vector irrep. From this we compute

$$\mathcal{Z} = \pi_F(\mathcal{E}) = \mathbb{Z}_2 \times \mathbb{Z}_2,\tag{3.55}$$

and the 0-form flavor symmetry group $\mathcal{F}$ is

$$\mathcal{F} = F/\mathcal{Z} = \frac{Sp(m_1) \times SO(4m_2) \times Sp(m_3)}{\mathbb{Z}_2},\tag{3.56}$$

where the $\mathbb{Z}_2$ in the denominator is the diagonal $\mathbb{Z}_2$ of the $\mathbb{Z}_2$ centers of $Sp(m_1)$, $SO(4m_2)$ and $Sp(m_3)$.

The groups $\Gamma^{(1)}$, $\mathcal{E}$ and $\mathcal{Z}$ sit in a short exact sequence (2.4) that becomes

$$0 \to \mathbb{Z}_2 \to \mathbb{Z}_4 \times \mathbb{Z}_2 \to \mathbb{Z}_2 \times \mathbb{Z}_2 \to 0.\tag{3.57}$$

This leads to a non-trivial 2-group symmetry with the Postnikov class

$$\Theta = \text{Bock}(w_2),\tag{3.58}$$

where $w_2$ is the obstruction class for lifting $\mathcal{F}$ bundles to $Sp(m_1) \times SO(4m_2) \times Sp(m_3)$ bundles. The Bockstein homomorphism appearing in (3.58) is associated to the short exact sequence

$$0 \to \mathbb{Z}_2 \to \mathbb{Z}_4 \to \mathbb{Z}_2 \to 0,\tag{3.59}$$

which is the non-split part of the short exact sequence (3.57). We can arrive at the same conclusions as above by using the charge matrix. For this, we write

$$Z_G = \mathbb{Z}_4^{\text{Spin}(4n_1+2)} \times \mathbb{Z}_2^{Sp(n_2)} \times \mathbb{Z}_4^{\text{Spin}(4n_3+2)},\tag{3.60}$$

and

$$Z_F = \mathbb{Z}_2^{Sp(m_1)} \times (\mathbb{Z}_2 \times \mathbb{Z}_2)^{\text{Spin}(4m_2)} \times \mathbb{Z}_2^{Sp(m_3)}.\tag{3.61}$$

The charge matrix can then be written as

$$\mathcal{M} = \begin{pmatrix} 4 & 0 & 0 & 0 & 0 & 0 & 0 & 2 & 2 & 0 & 0 & 0 & 1 & 1 \\ 0 & 2 & 0 & 0 & 0 & 0 & 0 & 0 & 1 & 1 & 1 & 0 & 0 & 0 \\ 0 & 0 & 4 & 0 & 0 & 0 & 0 & 0 & 0 & 0 & 2 & 2 & 1 & 3 \\ 0 & 0 & 0 & 2 & 0 & 0 & 0 & 1 & 0 & 0 & 0 & 0 & 0 & 0 \\ 0 & 0 & 0 & 0 & 2 & 0 & 0 & 0 & 0 & 1 & 0 & 0 & 1 & 0 \\ 0 & 0 & 0 & 0 & 0 & 2 & 0 & 0 & 0 & 1 & 0 & 0 & 0 & 1 \\ 0 & 0 & 0 & 0 & 0 & 0 & 2 & 0 & 0 & 0 & 0 & 1 & 0 & 0 \end{pmatrix}. \tag{3.62}$$

From this we compute

$$\begin{aligned} \Gamma^{(1)} &= \mathbb{Z}_2 \times \mathbb{Z}_1 \times \mathbb{Z}_1 \,, \\ \mathcal{E} &= \mathbb{Z}_4 \times \mathbb{Z}_1 \times \mathbb{Z}_2 \times \mathbb{Z}_1 \times \mathbb{Z}_1 \times \mathbb{Z}_1 \times \mathbb{Z}_1 \,, \end{aligned} \tag{3.63}$$

with

$$A_{\mathcal{E}}^{-1} = \begin{pmatrix} 1 & -2 & 0 & -2 & -2 & -4 & -2 \\ 0 & 1 & 0 & 1 & 1 & 2 & 1 \\ 0 & 0 & 1 & 0 & -1 & -1 & 0 \\ 0 & 0 & 0 & 1 & 1 & 2 & 1 \\ 0 & 0 & 0 & 0 & 1 & 1 & 0 \\ 0 & 0 & 0 & 0 & 0 & 1 & 0 \\ 0 & 0 & 0 & 0 & 0 & 0 & 1 \end{pmatrix}, \tag{3.64}$$

and

$$R^t = \begin{pmatrix} 2 & -1 & 0 & -1 & -1 & -2 & -1 \\ 0 & 1 & 0 & 1 & 1 & 2 & 1 \\ 0 & 0 & 2 & 0 & -1 & -1 & 0 \end{pmatrix}. \tag{3.65}$$

Thus, the generator of $\mathbb{Z}_2$ subfactor of $\Gamma^{(1)}$ embeds as twice the generator of the $\mathbb{Z}_4$ subfactor of $\mathcal{E}$. One can easily check that the two $\mathbb{Z}_1$ subfactors of $\Gamma^{(1)}$ do not have a non-trivial embedding into the $\mathbb{Z}_4$ or $\mathbb{Z}_2$ subfactor of $\mathcal{E}$.

Continuing, we find that

$$\mathcal{Z} = \mathbb{Z}_2 \times \mathbb{Z}_1 \times \mathbb{Z}_2 \times \mathbb{Z}_1 \times \mathbb{Z}_1 \times \mathbb{Z}_1 \times \mathbb{Z}_1 \,, \tag{3.66}$$

with $A_{\mathcal{Z}}$ being the identity matrix. Thus, the generator of the $\mathbb{Z}_4$ subfactor of $\mathcal{E}$ projects to the generator of the first $\mathbb{Z}_2$ subfactor of $\mathcal{Z}$, and the generator of the $\mathbb{Z}_2$ subfactor of $\mathcal{E}$ projects to the generator of the second $\mathbb{Z}_2$ subfactor of $\mathcal{Z}$. So far we have recovered the short exact sequence (3.57).

Now, we want to find the embedding of $\mathcal{Z}$ into $Z_F$ which would allow us to read $\mathcal{F}$ and the obstruction class $w_2$ appearing in the Postnikov class. For this we compute

$$\mathcal{M}_A[(A_{\mathcal{E}}^t)^{-1} Q^{-1} A_{\mathcal{Z}}^{-1}]_F = \begin{pmatrix} -1 & 2 & 0 & 2 & 0 & 0 & 0 \\ -1 & 2 & -1 & 2 & 2 & 0 & 0 \\ -2 & 4 & -1 & 4 & 2 & 2 & 0 \\ -1 & 2 & 0 & 2 & 0 & 0 & 2 \end{pmatrix}, \tag{3.67}$$

to find that the first $\mathbb{Z}_2$ subfactor of $\mathcal{Z}$ embeds as the diagonal of the first, second and fourth $\mathbb{Z}_2$ subfactors of $Z_F$, while the second $\mathbb{Z}_2$ subfactor of $\mathcal{Z}$ embeds as the diagonal of the second and third $\mathbb{Z}_2$ subfactors of $Z_F$. This confirms the result for the flavor symmetry group $\mathcal{F}$ in (3.56). The fact that the first $\mathbb{Z}_2$ subfactor of $\mathcal{Z}$ participates in the non-split part of (3.57) which embeds into $Z_F$ as above recovers the class $w_2$ appearing in (3.58).

## 3.6 Strings, 1-form Symmetries and Structure Groups

There is an equivalent way to see how the states given by the strings are consistent or not with the 1-form symmetry predicted by the low-energy gauge theory in 6d, which does not require knowing the charges of the states under the centers of the gauge and flavor symmetries. This method was proposed in [77], and relies on analyzing the Green-Schwarz-West-Sagnotti (GSWS) couplings present in the low-energy effective action in 6d, which are necessary for the cancellation of reducible gauge anomalies. A generic 6d theory has tensor multiplets $(\phi^i, t_2^i, \gamma_I^i)$ and vector multiplets $(A_\mu^i, \lambda_I^i)$, where $t_2^i$ are dynamical antisymmetric tensor fields, $I = 1, 2$ indicates an $SU(2)_R$ doublet, and $i$ is the index labelling the dynamical tensor multiplets.[13] The GSWS coupling reads

$$S_{\text{GSWS}} = 2\pi\Omega_{ij} \int_{M_6} t_2^i \wedge \frac{1}{4}\text{Tr}(F^j \wedge F^j), \tag{3.68}$$

where we only need the part related to the instanton density $I_4^j = \frac{1}{4}\text{Tr}(F^j \wedge F^j)$, and $\Omega_{ij}$ is the Dirac pairing in the string charge lattice. Due to tadpole cancellation, a non-trivial configuration $\int_{M_4 \subset M_6} I_4^j \in \mathbb{Z}$, where $M_4$ is a general submanifold of $M_6$, requires the presence of BPS strings whose induced charges are $Q^j = -\int_{M_4 \subset M_6} I_4^j \in \mathbb{Z}$. In addition Dirac quantisation asserts that

$$\langle Q^i, Q^j \rangle \equiv Q^i \Omega_{ij} Q^j \in \mathbb{Z}, \qquad \Omega_{ij} \in \mathbb{Z}, \qquad \forall i, j. \tag{3.69}$$

We now ask what happens when the $\frac{1}{4}\text{Tr}(F^j \wedge F^j)$ fractionalizes (the $\int_{M_4 \subset M_6} I_4^j$ is a fractional number) due to turning backgrounds that twist the gauge group by its center or subgroups thereof, i.e. bundles in $G/\Gamma^{(1)}$[14], where $\Gamma^{(1)} \subset Z_G$ (see table 2 for the list of centers $Z_G$). This means to activate a background field that is

$$B = w_2(G/\Gamma^{(1)}) \in H^2(BG/\Gamma^{(1)}, \Gamma^{(1)}), \tag{3.70}$$

where characteristic class $w_2$ is the obstruction of lifting a $G/\Gamma^{(1)}$ bundle to a $G$ bundle. For any $G$, which is not Spin(4N), the center is $Z_G = \mathbb{Z}_n$ and subgroups are given by $\Gamma^{(1)} = \mathbb{Z}_k$. Then we have that

$$\widetilde{B} = \frac{n}{k}B, \tag{3.71}$$

where $B$ is the background for $G/\Gamma^{(1)}$ and $\widetilde{B}$ for $G/Z_G$. The fractionalisation of $I_4$ then reads

$$I_4 = \frac{n^2 \alpha_G}{k^2} \mathfrak{P}(B) \mod \mathbb{Z}, \tag{3.72}$$

where $\mathfrak{P}(B)$ is the pontryagin square characteristic class and $\alpha_G$ encodes the fractionalisation of the instanton density, see table 2. The case of $G = \text{Spin}(4N)$ is slightly different, there are three different subgroups, $\mathbb{Z}_2^L$, $\mathbb{Z}_2^R$ and $\mathbb{Z}_2 \hookrightarrow \mathbb{Z}_2^L \times \mathbb{Z}_2^R$, which is the diagonal embedding. So in general we have.

$$\begin{aligned}
\Gamma^{(1)} = \mathbb{Z}_2^L : && I_4 &= \frac{N}{4}\mathfrak{P}(B_L) \mod \mathbb{Z}, \\
\Gamma^{(1)} = \mathbb{Z}_2^R : && I_4 &= \frac{N}{4}\mathfrak{P}(B_R) \mod \mathbb{Z}, \\
\Gamma^{(1)} = \mathbb{Z}_2 : && I_4 &= \frac{1}{2}B \cup B = \frac{1}{2}\mathfrak{P}(B) \mod \mathbb{Z},
\end{aligned} \tag{3.73}$$

where $B_L = B_R = B$.

---

[13]Note that the gauge group associated to a particular tensor labelled by $i$ can also be trivial.

[14]Where we suppressed the index $j$ for a moment.

Table 2: Center symmetries $Z_G$ and fractionalisation of the instanton density. For Spin($4N$) the two contributions consist of $\mathfrak{P}(B^{(L)} + B^{(R)})$ and $B^{(L)} \cup B^{(R)}$, respectively [89].

| $G$ | $Z_G$ | $\alpha_G$ |
|---|---|---|
| $SU(N)$ | $\mathbb{Z}_N$ | $\frac{N-1}{2N}$ |
| $Sp(N)$ | $\mathbb{Z}_2$ | $\frac{N}{4}$ |
| $\mathrm{Spin}(2N+1)$ | $\mathbb{Z}_2$ | $\frac{1}{2}$ |
| $\mathrm{Spin}(4N+2)$ | $\mathbb{Z}_4$ | $\frac{2N+1}{8}$ |
| $\mathrm{Spin}(4N)$ | $\mathbb{Z}_2 \times \mathbb{Z}_2$ | $\left(\frac{N}{4}, \frac{1}{2}\right)$ |
| $E_6$ | $\mathbb{Z}_3$ | $\frac{2}{3}$ |
| $E_7$ | $\mathbb{Z}_2$ | $\frac{3}{4}$ |

We now have all the ingredients and the fractionalisation, due to $G/\Gamma^{(1)}$ backgrounds, reads

$$S_{\text{GSWS}} = 2\pi\Omega_{ij} \int_{M_6} t_2^i \wedge \frac{n^2 \alpha_G^j}{k^2} \mathfrak{P}(B^j), \tag{3.74}$$

and Dirac quantisation for the induced charges on BPS strings [77] demands the following necessary condition,

$$Q_i = \Omega_{ij} \frac{n^2 \alpha_G^j}{k^2} \int_{M_4 \subset M_6} \mathfrak{P}(B^j) \in \mathbb{Z}, \qquad \forall i. \tag{3.75}$$

The first step of this procedure consist of turning on a background for the center symmetries which are compatible with the massless spectrum of the low-energy gauge theory in the tensor branch. Then with the above condition it is possible to understand whether the background is also consistent with the non-perturbative massive string states, which become massless for example in 6d SCFTs. This method was for example used in [77] to understand the fate of various 1-form symmetry in 6d. We show here in some explicit examples how it is possible to detect the quotient group $\mathcal{E}$ in the structure group and the consistency of the related background. This gives a hint towards the 2-group backgrounds. This is done by including the backgrounds for the center of various flavor symmetries. Let us look at a simple example

$$\overset{\mathfrak{so}(4N+2)}{4} \text{——} [\mathfrak{sp}(4N-6)] \tag{3.76}$$

First of all we can see how the strings are consistent with the $\Gamma^{(1)} = \mathbb{Z}_2 \subset \mathbb{Z}_4 = Z_{\mathrm{Spin}(4N+2)}$ 1-form symmetry, such that we have

$$\Omega_{11} \frac{n^2 \alpha_G^1}{k^2} = 16\alpha_{\mathrm{Spin}(4N+2)} \in \mathbb{Z}, \tag{3.77}$$

where $n = 4$, $k = 2$, $\alpha_{\mathrm{Spin}(4N+2)} = \frac{2N+1}{8}$ and $\Omega_{ij} = 4$. We can now activate general twisted backgrounds which fractionalise the instanton density including the flavor symmetries such that we have

$$S_{\text{GSWS}} = 2\pi \int t_2 \wedge \left(\frac{2N+1}{2}\mathfrak{P}[w_2(\mathrm{Spin}(4N+2)/\mathbb{Z}_4)] - \frac{2N-3}{2}\mathfrak{P}[w_2(Sp(4N-6)/\mathbb{Z}_2)]\right). \tag{3.78}$$

We see that the most general choice of background that is compatible with (3.75) is

$$w_2(\text{Spin}(4N+2)/\mathbb{Z}_4) = 2B - w_2(Sp(4N-6)/\mathbb{Z}_2) \quad \text{mod} \quad 4, \tag{3.79}$$

where we recall that $B$ is the background field for the 1-form symmetry.[15] From this we gain several pieces of information. First, (3.79) is just the manifestation of $\mathcal{E} = \mathbb{Z}_4$. Secondly, $w_2(Sp(4N-6)/\mathbb{Z}_2)$ is allowed and therefore the flavor symmetry is $\mathcal{F} = Sp(4N-6)/\mathbb{Z}_2$. Finally, the form of the (3.79) hints also at the two group symmetry, where the $\mathbb{Z}_2$ 1-form symmetry and the $\mathbb{Z}_2$ quotient of the flavor symmetry mix to give a non-trivial element in $\mathbb{Z}_4$.

A second illustrative example is provided by just attaching an E-string to this theory, specializing to $N = 2$:

$$[\mathfrak{su}(4)] \underline{\hspace{3cm}} \overset{\emptyset}{1} \underline{\hspace{3cm}} \overset{\mathfrak{so}(10)}{4} \underline{\hspace{2cm}} [\mathfrak{sp}(2)] \quad , \tag{3.80}$$

and therefore the intersection pairing in the string lattice is

$$\Omega = \begin{pmatrix} 1 & -1 \\ -1 & 4 \end{pmatrix}. \tag{3.81}$$

This case does not have a 1-form symmetry since the string states break the one predicted just from the gauge theory massless spectrum as one can see from,

$$S_{\text{GSWS}} = -2\pi \int t_2^1 \wedge \left( \frac{5}{8} \mathfrak{P}[w_2(\text{Spin}(10)/\mathbb{Z}_4)] \right), \tag{3.82}$$

where $t_2^1$ is the tensor which charges the E-string. From this we can see that there is no subgroup of the center $\mathbb{Z}_4$ satisfied integrality of the induced gauge charges. This is just the low-energy manifestation that the E-strings states transforming under the spinor representations of Spin(10). For instance, the fractionalized GSWS coupling (3.82) can be thought as generated by integrating out massive E-string states, when going from the SCFT in the UV to the low-energy theory in the tensor branch. Having now 1-form symmetry implies that we do not have a 2-group. On the other hand we can still activate twisted backgrounds for the flavor symmetries compatible with the low-energy massless matter, and understand what are the backgrounds allowed by the BPS strings. In this case we get two conditions,

$$
\begin{aligned}
S_{\text{GSWS}_1} &= -2\pi \int t_2^1 \wedge \left( \frac{5}{8} \mathfrak{P}[w_2(\text{Spin}(10)/\mathbb{Z}_4) + \frac{3}{8} \mathfrak{P}[w_2(SU(4)/\mathbb{Z}_4)] \right), \\
S_{\text{GSWS}_2} &= +2\pi \int t_2^2 \wedge \left( \frac{5}{2} \mathfrak{P}[w_2(\text{Spin}(10)/\mathbb{Z}_4) - \frac{1}{2} \mathfrak{P}[w_2(Sp(2)/\mathbb{Z}_2)] \right),
\end{aligned}
\tag{3.83}
$$

where $t_2^2$ is the tensor with self-charge 4. The integrality for the second line is satisfied when (3.79) holds, where $B$ now is simply the background for a $\mathbb{Z}_2 \subset \mathbb{Z}_4 = Z(\text{Spin}(10))$, not related to any 1-form symmetry. For the induced charges on the E-string to be integral we need the integrality of quantity which multiplies the $t_2^1$ in the first line of (3.83). The most general choice which satisfies this is given by

$$w_2(SU(4)/\mathbb{Z}_4) = w_2(\text{Spin}(10)/\mathbb{Z}_4) = 2B - w_2(Sp(4N-6)/\mathbb{Z}_2) \quad \text{mod} \quad 4. \tag{3.84}$$

This implies that $\mathcal{E} = \mathbb{Z}_4$ in the structure group and the full flavor symmetry of the 6d theory is

$$\mathcal{F} = \frac{SU(4) \times Sp(2)}{\mathbb{Z}_4}, \tag{3.85}$$

where $\mathbb{Z}_2 \subset \mathbb{Z}_4$ does not act on $Sp(2)$.

---

[15]One can check this by expanding the pontryagin square in cup products [13].

# 4 Abelian and Discrete 0-Form Symmetries

Some of the theories we encountered in the discussion of 2-groups in 6d SCFTs have abelian flavor symmetries. These can be broken by ABJ anomalies. In addition we discuss a mixed anomaly between 0-form and 1-form symmetries that can exist in such theories.

## 4.1 ABJ Anomaly

In order to know the full flavor symmetry of 6d field theories we need to understand the abelian components. In particular, not all the $U(1)$ symmetries which can be seen from the lagrangian will survive quantum mechanically. This is due to the presence of ABJ anomalies [90], and holographically in [91]. Let us consider a 6d theory in the tensor branch with a certain number of abelian flavor symmetries labelled by $U(1)_\ell$ and with gauge vector multiplets transforming in the adjoint representation of $\mathfrak{g}_i$. The matter is charged under $U(1)_\ell$ with charge $q_\ell$ and transforms in a representation $\rho(\mathfrak{g}_i)$. By evaluating 1-loop diagrams there is an ABJ anomaly in 6d, and its anomaly polynomial reads,[16]

$$I_{\text{ABJ}} = \sum_{\text{matter}} q_\ell F_{U(1)_\ell} \frac{1}{6} \text{tr}_\rho(F^3_{\mathfrak{g}_i}) = \sum_{\text{matter}} q_\ell F_{U(1)_\ell} \frac{A(\rho_i)}{6} \text{Tr}_{\text{fund}}(F^3_{\mathfrak{g}_i}), \tag{4.1}$$

where $A(\rho)$ is called the anomaly coefficient, which normalises the cubic trace of a representation in terms of the fundamental representation, which has $A(\text{fund}) = 1$. Notice also that the cubic trace is non-vanishing only for $\mathfrak{g} = \mathfrak{su}$. Moreover, we have that $\frac{1}{6}\text{Tr}_{\text{fund}}(F^3_{\mathfrak{su}(N)}) = \frac{c_3}{2}$, where $\frac{c_3}{2}$ is the third Chern class. For an $SU(N)$ bundle, $\frac{c_3}{2}$ is always integer on a compact 6-manifold with an almost complex structure[17], that is necessary to define Chern classes, see appendix C.

A first consequence of the ABJ anomaly is that under a $U(1)_\ell$ rotation, one can always choose the $\theta = 0$ in a $\theta$-angle term like $\mathcal{L} \supset \frac{\theta}{6}\text{Tr}_{\text{fund}}(F^3_{\mathfrak{g}_i})$ as long as we do not have $SU(3)$ NHCs participating in the 6d theory under consideration. Crucially there are $U(1)$ combinations that lead to a vanishing ABJ anomaly. These $U(1)$ symmetries survive as quantum symmetries of the theory, which in total are $U(1)^{\#(\text{lagrangian } U(1)s) - \#(SU \text{ gauge nodes})}$. In addition there can be discrete unbroken transformations which form the discrete part 0-form symmetry group, that is $\Gamma^{(0)}_{\text{tor}} \subset \Gamma^{(0)} \subset \prod_\ell U(1)_\ell$. Both the continuous and torsion part of $\Gamma^{(0)}$ can be read off from a basis change which preserves the lattice of $U(1)_\ell$ charges, $\widetilde{F}_{U(1)_\ell} = \Lambda_{\ell\ell'} F_{U(1)_{\ell'}}$ such that $\Lambda_{\ell\ell'}$ is a matrix with unit determinant. Moreover, $\widetilde{F}_{U(1)_i}$ are the combinations that appear in front of the $\text{Tr}_{\text{fund}}(F^3_{\mathfrak{g}_i})$ with coefficients,

$$\sum_{\text{matter}} q_\ell F_{U(1)_\ell} A(\rho_i) = p_i \widetilde{F}_{U(1)_i}. \tag{4.2}$$

The torsional backgrounds such that $p_i \widetilde{F}_{U(1)_i} = 0$ exactly define

$$\Gamma^{(0)}_{\text{tor}} = \prod_i \mathbb{Z}_{p_i}. \tag{4.3}$$

The full abelian symmetry reads,

$$\Gamma^{(0)} = U(1)^{\#(\text{lagrangian } U(1)s) - \#(SU \text{ gauge nodes})} \prod_i \mathbb{Z}_{p_i}. \tag{4.4}$$

---

[16]Alternatively to derive the ABJ anomaly one can take the $\frac{\text{Tr}_{\rho(\mathfrak{h})}F^4}{24}$ hypermultiplet contribution, see appendix A of [92], and decompose the $\mathfrak{h} \subset \mathfrak{g} \oplus \mathfrak{u}(1)$, where $\mathfrak{g}$ is non-abelian. We also recall that in 6d the hypermultiplet contains a single Weyl fermion which transforms in a doublet of $SU(2)_R$.

[17]A 6-dimensional manifold has a almost complex structure if and only if has a spin$^c$ structure. We restrict here to $M_6$ with a spin$^c$ structure.

We now illustrate this in an explicit example. Let us take the following type 5 example,

$$
\begin{array}{ccc}
\underset{2}{\mathfrak{su}(12)} & \underline{\hspace{2cm}} & \underset{2}{\mathfrak{so}(22)} \\
| & & | \\
[\mathfrak{u}(2)] & & [\mathfrak{sp}(2)]
\end{array}
\tag{4.5}
$$

In this quiver there are two continuous abelian symmetries from the lagrangian. The first one is $\mathfrak{u}(1)_f \subset \mathfrak{u}(2)$, the second is given by the baryonic symmetry rotating the hypers between $\mathfrak{su}(12)$ and $\mathfrak{so}(22)$, which we denote by $\mathfrak{u}(1)_b$. Their ABJ anomaly reads,

$$
I_{\text{ABJ}} = \left( -2F_{U(1)_f} + 22F_{U(1)_b} \right) \frac{1}{6} \text{Tr}_{\text{fund}}(F^3_{\mathfrak{su}(12)}).
\tag{4.6}
$$

We can see that there is a combination of the two $U(1)$s which is free from ABJ anomalies and remains a symmetry of the quantum theory. Upon the following lattice of charge preserving change of basis,

$$
\begin{pmatrix} \widetilde{F}_{U(1)_1} \\ \widetilde{F}_{U(1)_2} \end{pmatrix} = \begin{pmatrix} -1 & 11 \\ 0 & 1 \end{pmatrix} = \begin{pmatrix} F_{U(1)_f} \\ F_{U(1)_b} \end{pmatrix},
\tag{4.7}
$$

the ABJ anomaly now reads,

$$
I_{\text{ABJ}} = 2\widetilde{F}_{U(1)_1} \frac{1}{6} \text{Tr}_{\text{fund}}(F^3_{\mathfrak{su}(12)}).
\tag{4.8}
$$

This means that the continuous anomaly free combination is given by $\widetilde{F}_{U(1)_1} = -F_{U(1)_f} + 11F_{U(1)_b} = 0$, and that torsional background configurations, such that

$$
2\widetilde{F}_{U(1)_1} = 0,
\tag{4.9}
$$

are not anomalous, leading to $\Gamma^{(0)}_{\text{tor}} = \mathbb{Z}_2$. The full abelian symmetry is

$$
\Gamma^{(0)} = U(1) \times \mathbb{Z}_2.
\tag{4.10}
$$

Let us consider another illustrative example,

$$
[\mathbf{\Lambda}^2] \underline{\hspace{2cm}} \underset{0}{\mathfrak{su}(4)} \underline{\hspace{2cm}} [\mathbf{S}^2],
\tag{4.11}
$$

where the flavor symmetry algebra rotating the two-index antisymmetric $\mathbf{\Lambda}^2$ is $\mathfrak{sp}(1)$ and the one associated to the two-index symmetric $\mathbf{S}^2$ is $\mathfrak{so}(2)_{\mathbf{S}^2} = \mathfrak{u}(1)_{\mathbf{S}^2}$. So we have an abelian flavor symmetry and its ABJ anomaly is,

$$
I_{\text{ABJ}} = 8F_{U(1)_{\mathbf{S}^2}} \frac{1}{6} \text{Tr}_{\text{fund}}(F^3_{\mathfrak{su}(4)}),
\tag{4.12}
$$

where $A(\mathbf{S}^2(\mathfrak{su}(4))) = 8$. In this case there is no continuous ABJ anomaly free combination, but there is a discrete remnant given by $8F_{U(1)_{\mathbf{S}^2}} = 0$, that gives

$$
\Gamma^{(0)} = \mathbb{Z}_8.
\tag{4.13}
$$

## 4.2 A New Mixed Anomaly Between Flavor and 1-Form Symmetries

We now consider the situation, when turning on the 1-form symmetry backgrounds, where $\frac{1}{6}\mathrm{Tr}_{\mathrm{fund}}(F^3_{\mathfrak{g}_i})$ fractionalizes. In particular the cases that appear in this paper are such that $\mathfrak{g} = \mathfrak{su}(2n)$ and the one-form symmetry, which is a subgroup of the center $\mathbb{Z}_{2n}$ of $SU(2n)$ is $\Gamma^{(0)} = \mathbb{Z}_2$. It is then possible to rewrite the cubic trace as follows

$$\frac{1}{6}\mathrm{Tr}_{\mathrm{fund}}(F^3_{\mathfrak{su}(2n)}) = \frac{1}{2}c_3(F'_{\mathfrak{u}(2n)}) - \left(\frac{(2n)}{2} - 1\right)B_2 c_2(F'_{\mathfrak{u}(2n)}) + \frac{(2n)((2n)-1)((2n)-2)}{6}B_2^3, \tag{4.14}$$

where $B_2$ is the 1-form symmetry, $\Gamma^{(1)} = \mathbb{Z}_2$, background with $\oint B_2 \in \frac{\mathbb{Z}}{2}$ periods, and the $\mathfrak{u}_{2n}$ bundle is related to the $\mathfrak{su}_{2n}$ by

$$A'_{\mathfrak{u}_{2n}} = A_{\mathfrak{su}_{2n}} + \frac{1}{2n}\frac{2n}{2}B\mathbb{I}_{2n}, \tag{4.15}$$

where $2B_2 = dB$, and the $U(1)$ field is reabsorbed by the 1-form symmetry transformation $A'_{\mathfrak{u}(2n)} \to A'_{\mathfrak{u}_N} + \mathbb{I}_{2n}\lambda$, $B_2 \to B_2 + d\lambda$. The first two terms in (4.14) are integer valued and do not lead to any anomaly. This is because $\int c_3 \in 2\mathbb{Z}$ for a $\mathfrak{u}(2n)$ vector bundle on a compact 6-manifold with an almost complex structure, see appendix C. The second term vanishes mod $\mathbb{Z}$. All in all, the result of performing a $\Gamma^{(0)}$ transformation leads to following mixed 't Hooft anomaly,

$$\mathcal{A} = \frac{2n(2n-1)(2n-2)}{6}p_i \sum_i a_i B_2^3, \tag{4.16}$$

where $a_i$ have discrete periods, $\oint a_i \in \frac{\mathbb{Z}}{p_i}$, and $\oint B_2 \in \frac{\mathbb{Z}}{2}$ periods. We can see that for (4.5), the anomaly on a general 6-manifold[18] reads,

$$\mathcal{A} = \frac{11 \times 5}{2}\tilde{a}\tilde{B}_2^3 \qquad \mathrm{mod} \qquad \mathbb{Z}, \tag{4.17}$$

where $\tilde{a}$ and $\tilde{B}_2$ have integer periods mod 2. For (4.11) we also have that on a 6-general manifold,

$$\mathcal{A} = \frac{1}{2}\tilde{a}\tilde{B}_2^3 \qquad \mathrm{mod} \qquad \mathbb{Z}, \tag{4.18}$$

where $\tilde{a}$ has integer periods mod 8 and $\tilde{B}_2$ has integer periods mod 2.

There is a potential clash between the existence of the above anomaly $\mathcal{A}$ and the existence of 2-group symmetry, if the same 1-form symmetry participates in both. The existence of 2-group implies that $\delta B \neq 0$, which forces $\delta \mathcal{A} \neq 0$ making the expression $\mathcal{A}$ for the anomaly ill-defined. Merrily, such a clash does not occur for 6d SCFTs, at least for the type of 2-group symmetry being discussed in this paper. The reason for this is that, from the analysis of section 5, we know that 1-form symmetry participating in 2-group symmetry does not have non-trivial projection on the center of any $SU$ gauge group appearing on the tensor branch of the theory. On the other hand, the above anomaly arises only for 1-form symmetries that have a non-trivial projection on some $SU$ gauge group.

## 5 Proof of the Classification of 6d Theories With 2-Group Symmetries

In this section, we classify 6d SCFTs and LSTs with non-trivial 2-group symmetries. The output of this section is a list of building blocks for theories that can have non-trivial 2-group symmetries, which are listed in section 5.1.

---

[18]i.e. with no condition on the spin structure, or on its pontryagin classes.

## 5.1 Building Blocks For 6d Theories with 2-Group Symmetries

From the analysis of the subsequent section, we find that the 6d theories admitting 2-group symmetries (of the type being studied in this paper, see section 2.1) are obtained by composing the following building blocks:

**Block 1**

$$- - - - - - - - \mathfrak{so}(4n_1+2) \longrightarrow \mathfrak{sp}(n_2) \longrightarrow \mathfrak{so}(4n_3+2) - - - - - - - - . \qquad (5.1)$$

The dashed lines represent series of alternating $\mathfrak{so}(4N+2) - \mathfrak{sp}(M)$ gauge algebras.

**Block 2**

$$- - - - - - - - \mathfrak{so}(4n_1+2) \longrightarrow \mathfrak{sp}(n_2) \longrightarrow \mathfrak{so}(4n_3+2) - - - - - - - -$$

$$\mathfrak{so}(4m_1) \cdots\cdots\cdots\cdots\cdots\cdots\cdots \qquad (5.2)$$

The dashed lines represent series of alternating $\mathfrak{so}(4N+2) - \mathfrak{sp}(M)$ gauge algebras. The dotted line represents a series of alternating $\mathfrak{so}(4N) - \mathfrak{sp}(M)$ gauge algebras.

**Block 3**

$$- - - - - - - - \mathfrak{so}(4n_1+2) \longrightarrow \mathfrak{su}(n_2) \cdots\cdots\cdots\cdots . \qquad (5.3)$$

The dashed line represents a series of alternating $\mathfrak{so}(4N+2) - \mathfrak{sp}(M)$ gauge algebras. The dotted line represents a series of $\mathfrak{su}(P)$ gauge algebras.

Combining these building blocks with the imposition of rank constraints, and ensuring that an $\mathfrak{sp}$ node always have at least two non-flavor neighboring nodes (which is required for the existence of 1-form symmetry participating in 2-group), we are lead to a full list of 6d theories admitting 2-group symmetries of the type studied in this paper. These theories are discussed in detail in section 3.3.

## 5.2 Proof Strategy

Let us begin by setting up some notation first. Let $i$ parametrize different non-flavor nodes and let $k_i$ be the value of the node $i$. Let $\mathfrak{g}_i$ be the gauge algebra carried by the node $i$, and $G_i$ be the simply-connected associated to $\mathfrak{g}_i$.

For the purposes of deduction of 2-group symmetries, we can choose the gauge group to be $G = \prod_i G_i$. A different choice $G'$ of gauge group is obtained by gauging a subgroup $\Gamma^{(1)'}$ of the 1-form symmetry group $\Gamma^{(1)}$. The theory with gauge group $G'$ carries an $\Gamma^{(1)}/\Gamma^{(1)'}$ 1-form symmetry and a potential 2-group symmetry whose Postnikov class $\Theta'$ is given by

$$\Theta' = \pi'(\Theta), \qquad (5.4)$$

where $\pi'$ is the natural map

$$\pi' : H^3(B\mathcal{F}, \Gamma^{(1)}) \to H^3(B\mathcal{F}, \Gamma^{(1)}/\Gamma^{(1)'}), \qquad (5.5)$$

induced by the map $\Gamma^{(1)} \to \Gamma^{(1)}/\Gamma^{(1)'}$. We can also write

$$\Theta' = \text{Bock}'(w_2) + \cdots, \qquad (5.6)$$

where $w_2 \in H^2(B\mathcal{F}, \mathcal{Z})$ is again the obstruction class for lifting $\mathcal{F}$ bundles to $F$ bundles, and Bock$'$ is the Bockstein homomorphism induced by the short exact sequence

$$0 \to \Gamma^{(1)}/\Gamma^{(1)'} \to \mathcal{E}/\Gamma^{(1)'} \to \mathcal{Z} \to 0 \,. \tag{5.7}$$

In particular, we have

$$\text{Bock}'(w_2) = \pi'(\text{Bock}(w_2)) \,. \tag{5.8}$$

Thus, if $\text{Bock}'(w_2) \neq 0$, then we must have $\text{Bock}(w_2) \neq 0$. This means that 2-group symmetry (of the type studied in this paper) for theory with gauge group $G'$ can be completely understood if one understands 2-group symmetry (of the type studied in this paper) for the theory with gauge group $G$. Consequently, in the rest of the classification, we will assume that the gauge group is $G = \prod_i G_i$. At the end of the classification, we will study all the possible 1-form symmetry gaugings that lead to theories with other gauge groups that also carry 2-group symmetries.

Let $Z_i$ be the center of the group $G_i$. Then $Z_G = \prod_i Z_i$. Let $\pi_i : Z_G \times Z_F \to Z_i$ be the projection map onto $Z_i$. Let us also decompose the group $F$ into its factors $F_a$. There are various allowed possibilities for $F_a$:

- Continuous and non-abelian. In this case it is a localized flavor symmetry.

- Continuous and abelian. In this case it can be localized or delocalized.

- Finite and abelian. In this case it is delocalized, and arises from remnant of a continuous abelian flavor symmetry afflicted by ABJ anomaly.

Let $Z_a$ be the center of $F_a$ and let $\pi_a : Z_G \times Z_F \to Z_a$ be the projection map onto $Z_a$. For the rest of this section, we study the consequences of $\mathcal{E}$ containing an element $\alpha$ such that

- $\alpha \neq p\alpha'$ for $p > 1$ and $\alpha' \in \mathcal{E}$.

- $\alpha$ generates a $\mathbb{Z}_n$ subgroup of $\mathcal{E}$.

- $\pi_F(\alpha)$ generates a $\mathbb{Z}_k$ subgroup of $\mathcal{Z}$.

In such a situation $\alpha$ generates a piece of (2.4) of the form

$$0 \to \mathbb{Z}_{n/k} \to \mathbb{Z}_n \to \mathbb{Z}_k \to 0 \,, \tag{5.9}$$

and provides a contribution of the form

$$\Theta = \text{Bock}(w_2) + \cdots \,, \tag{5.10}$$

to the 2-group symmetry, where $w_2 \in H^2\left(B\frac{F}{\mathbb{Z}_k}, \mathbb{Z}_k\right)$ is the obstruction class for lifting $F/\mathbb{Z}_k$ bundles to $F$ bundles. The non-triviality of $\text{Bock}(w_2)$ requires $gcd(k, n/k) \geq 2$, which implies $k \geq 2$ and $n \geq 4$.

Let us also define

$$\beta := k\alpha \,, \tag{5.11}$$

which has the property $\pi_F(\beta) = 0$, and hence generates the $\mathbb{Z}_{n/k}$ 1-form symmetry appearing in (5.9).

The rest of this section is organized as follows. In section 5.3, we first argue that only continuous non-abelian flavor symmetries participate in 2-groups of the type discussed in this paper. From section 5.4 onward, we begin exploring the consequences of the existence of the element $\alpha$ discussed above. We define the notion of a special node, which is a non-flavour node where $\alpha$ is represented faithfully. A theory exhibiting 2-group symmetry must contain at

least one special node. We find that the special node can either carry an $SU(N)$ gauge group or a Spin$(4M+2)$ gauge group. In section 5.4, we study all theories containing a special node of $SU$ type, and find that no such theory can have 2-group symmetry. In section 5.5, we study all theories containing a special node of Spin type. We find many building blocks consistent with 2-group symmetry that can be composed to build theories having 2-group symmetries. These building blocks are collected in section 5.1. The list of theories appearing in section 3.3 is obtained by composing these building blocks.

## 5.3 Removing Abelian Factors Inside $F$

Consider first a situation such that an $F_a = U(1)$ participates in (5.9), i.e. we have $\pi_a(\alpha) \neq 1 \in Z_a = U(1)$. Furthermore, we can choose $F_a = U(1)$ to be large enough that $\pi_a(\alpha)$ generates a $\mathbb{Z}_k$ subgroup of $Z_a = U(1)$. Then, we can express the contribution (5.10) as

$$\Theta = \text{Bock}(w_2) + \cdots, \tag{5.12}$$

with $w_2 \in H^2\left(B\frac{F_a}{\mathbb{Z}_k}, \mathbb{Z}_k\right)$ being the obstruction class for lifting $F_a/\mathbb{Z}_k = U(1)/Z_k$ bundles to $F_a = U(1)$ bundles.

This description of $w_2$ and Bock$(w_2)$ makes it manifest that Bock$(w_2) = 0$. To see this, notice that we can identify

$$w_2 = c_1 \pmod{k}, \tag{5.13}$$

with $c_1$ being the first Chern class of $F_a/\mathbb{Z}_k = U(1)/\mathbb{Z}_k \simeq U(1)$ bundles. This makes it clear that $w_2$ is in the image of the map

$$H^2\left(B\frac{F_a}{\mathbb{Z}_k}, \mathbb{Z}_n\right) \to H^2\left(B\frac{F_a}{\mathbb{Z}_k}, \mathbb{Z}_k\right), \tag{5.14}$$

associated to the map $\mathbb{Z}_n \to \mathbb{Z}_k$ in (5.9), since it is the image of $c_1 \pmod n \in H^2\left(B\frac{F_a}{\mathbb{Z}_k}, \mathbb{Z}_n\right)$. By exactness of long exact sequence in cohomology, this implies that $w_2$ is in the kernel of the connecting Bockstein homomorphism

$$H^2\left(B\frac{F_a}{\mathbb{Z}_k}, \mathbb{Z}_k\right) \to H^3\left(B\frac{F_a}{\mathbb{Z}_k}, \mathbb{Z}_{n/k}\right). \tag{5.15}$$

Thus, we can discard continuous abelian $F_a$ from $F$ as far as deduction of non-trivial 2-group symmetries is concerned.

Now, consider a situation such that an $F_a = \mathbb{Z}_m$ participates in (5.9), i.e. we have $\pi_a(\alpha) \neq 1 \in Z_a = \mathbb{Z}_m$. Furthermore, we can choose $F_a = \mathbb{Z}_m$ to be large enough that $\pi_a(\alpha)$ generates a $\mathbb{Z}_k$ subgroup of $Z_a = \mathbb{Z}_m$. Then, we can express the contribution (5.10) as

$$\Theta = \text{Bock}(w_2) + \cdots, \tag{5.16}$$

with $w_2 \in H^2\left(B\frac{F_a}{\mathbb{Z}_k}, \mathbb{Z}_k\right)$ being the obstruction class for lifting $F_a/\mathbb{Z}_k = \mathbb{Z}_{m/k}$ bundles to $F_a = \mathbb{Z}_m$ bundles.

We can now show that Bock$(w_2) = 0$. Consider $w_2' \in H^2\left(B\frac{F_a}{\mathbb{Z}_k}, \mathbb{Z}_n\right)$ describing the obstruction of lifting $F_a/\mathbb{Z}_k = \mathbb{Z}_{m/k}$ bundles to $\mathbb{Z}_{mn/k}$ bundles. We can recognize $w_2$ as the image of $w_2'$ under the map (5.14). By the same argument as above, this shows that Bock$(w_2) = 0$.

Thus, we can discard both continuous and finite abelian $F_a$ from $F$ as far as deduction of non-trivial 2-group symmetries is concerned.

## 5.4 Rejecting The Possibility of Special Node of Type $SU$

We must have at least one non-flavor node $i$ such that $\pi_i(\alpha)$ generates a $\mathbb{Z}_n$ subgroup of $Z_i$. This means that $Z_i$ must contain an element of order at least 4, since $n \geq 4$. This restricts the possible values of $G_i$ to be either $\mathrm{Spin}(4M+2)$ or $SU(N)$ with $N \geq 4$. We will call such a node $i$ a *special node*. Note that there can be multiple special nodes in a 6d theory. If a special node carries $SU(N)$ gauge group for some $N$, we call it a special node of $SU$ type. If a special node carries $\mathrm{Spin}(4M+2)$ gauge group for some $M$, we call it a special node of Spin type.

Let us begin by considering a theory that admits a special node $i$ of $SU$ type carrying $G_i = SU(N)$. If any other node $j$ carrying $G_j = SU(M)$ is a neighbor of $i$, then we must have a bifundamental in between them. For this bifundamental to be left invariant under $\alpha$, $\pi_j(\alpha)$ must generate a $\mathbb{Z}_n$ subgroup inside $Z_j$. Thus, the node $j$ is also a special node of type $SU$.

In general, let us define $\mathcal{I}$ to be the set of nodes carrying $SU$ gauge groups that can be connected to the special node $i$ by a chain of $SU$ gauge nodes. Any node $j \in \mathcal{I}$ is also a special node of $SU$ type.

Let us now assume that there is a node $k$ carrying non-$SU$ gauge group which is a neighbor of a node $j \in \mathcal{I}$:

$$\boxed{G_i = SU(N)} \quad\rule{1cm}{0.4pt}\quad \boxed{SU(M)} \;\text{-----}\; \boxed{\mathcal{I} \ni G_j = SU(P)} \quad\rule{1cm}{0.4pt}\quad \boxed{G_k = ?, G_k \neq SU} \tag{5.17}$$

We first consider gauge-theoretic options for the node $k$:

- $G_k = Sp(Q)$. In this case there must be a bifundamental hyper between $k$ and $j$. This hyper cannot be invariant under $\alpha$ as $Z_k = \mathbb{Z}_2$ can only act with order $\leq 2$ on the bifundamental, while $\pi_j(\alpha)$ acts with order $n \geq 4$ on it.

- $G_k = \mathrm{Spin}(2Q)$. In this case there must be a bifundamental hyper between $k$ and $j$. $\pi_j(\alpha)$ and $\pi_j(\beta)$ act non-trivially on the bifundamental, but it is not possible for both $\pi_k(\alpha)$ and $\pi_k(\beta)$ to act non-trivially on the bifundamental, since the subgroup of $Z_k$ acting faithfully on the bifundamental is only $\mathbb{Z}_2$. This is in contradiction with the presence of 2-group symmetry.

- $G_k = \mathrm{Spin}(2Q+1)$ or $G_2$. In this case the hyper between $k$ and $j$ transforms as $\boldsymbol{F} \otimes R$ of $G_j \times G_k$ where $R$ is an irrep of $G_k$. Since $Z_k \leq \mathbb{Z}_2$, the same argument as for the $G_k = Sp(Q)$ case above removes these possibilities for $G_k$ as well.

We therefore cannot have a non-$SU$ gauge-theoretic neighbour of $\mathcal{I}$. Let us now consider non-gauge-theoretic neighbors $k$ of a node $j \in \mathcal{I}$:

- $G_k = SU(1)$. In this case we must have $G_j = SU(2)$, but that is not possible since $j$ is a special node and hence $P \geq 4$.

- $G_k = Sp(0)$ is possible only if $G_j = SU(P \leq 9)$. For $G_j = SU(9)$, there is no flavor symmetry arising from the $Sp(0)$ node, and the BPS string arising from the $Sp(0)$ node contributes a state charged as $\boldsymbol{\Lambda}^3$ of $SU(9)$. This has no direct contradiction with 2-group symmetry and therefore provides a consistent ingredient to build models of this class that have 2-group symmetries.

- For $G_j = SU(8)$ and $G_k = Sp(0)_\pi$, there is only a $\mathfrak{u}(1)$ flavor symmetry arising from the $Sp(0)$ node, and the BPS string arising from the $Sp(0)$ node contributes a state charged as $\boldsymbol{F}$ of $SU(8)$. This is not consistent with 2-group symmetry.

- For $G_j = SU(8)$ and $G_k = Sp(0)_0$, there is an $\mathfrak{su}(2)$ flavor symmetry arising from the $Sp(0)$ node, and the BPS string arising from the $Sp(0)$ node contributes two states: one charged as $\mathbf{\Lambda}^2$ of $SU(8)$, and the other charged as $\mathbf{F}$ of $SU(2)$. Now, $\alpha$ must act nontrivially on the $\mathbf{\Lambda}^2$ string state, but there is no flavor symmetry associated to it that can compensate this action. This is in contradiction with 2-group symmetry.

- $\mathcal{G}_j = SU(P \leq 7)$ with $P \neq 4$. The center $Z_j$ is such that $j$ cannot be chosen as the special node $i$. So we need not consider these possibilities.

- $\mathcal{G}_j = SU(4)$. There is an $\mathfrak{so}(10)$ flavor symmetry arising from the $Sp(0)$ node, and the BPS string arising from the $Sp(0)$ node contributes a state charged as $\mathbf{F} \otimes \mathbf{S}$ of $SU(4) \times \mathrm{Spin}(10)$. This has no direct contradiction with 2-group symmetry, and is therefore a consistent ingredient.

Combining the ingredients found above, there is only one configuration of nodes that can sit inside a model with a special node $i$ of $SU$ type that is consistent with 2-group symmetry:

$$\mathbf{\Lambda}^2 \text{——} SU \text{——} SU \text{- - - -} SU \text{——} \mathbf{S}^2 \ . \tag{5.18}$$

This follows by analyzing the constraints on the ranks of the gauge groups, and requiring that all fundamentals of $SU$ gauge nodes are gauged by other gauge groups (for $\beta$ to generate a non-trivial 1-form symmetry). Notice that the above configuration is already an LST, so no further nodes can be attached to it without ruining consistency.

Unfortunately, this model does not have 2-group symmetry as we must have $\pi_a(\alpha) \neq 1 \in F_a = U(1)$ flavor symmetry rotating $\mathbf{S}^2$. But, as we saw in the previous subsection, the associated Postnikov class must be trivial in such a situation.

## 5.5 Constraining The Possible Theories Carrying Special Nodes of Type Spin

Now consider theories which contain a special node $i$ of Spin type. For such a theory, $\beta$ must be of order two, and $\alpha$ must be of order four. The value of the node must be $k_i = 4$, otherwise $\beta$ is not a part of 1-form symmetry.

We start by understanding the possible gauge-theoretic blocks neighboring the node $i$. Consider a node $j$ which is a neighbor of $i$, carrying $G_j = Sp(n)$. It gives rise to a bifundamental half-hyper of $G_i \times G_j$ and, more importantly, the string associated to the node $j$ gives rise to a state charged as $\mathbf{S}$ of $G_i$. Thus, $j$ must have another neighbor $k$ under which the string state is charged, leading to a sub-graph of the form:

$$\boxed{G_i = \mathrm{Spin}(4M+2)} \text{——} \boxed{G_j = Sp(N)} \text{——} \boxed{G_k = ?} \ . \tag{5.19}$$

There are various possibilities to consider for the node $k$:

- $G_k = \mathrm{Spin}(4P+2)$. Now the string state associated to $j$ is charged as $\mathbf{S} \otimes \mathbf{S}$ of $G_i \times G_k$. Invariance of this state under $\alpha$ leads us to the conclusion that $k$ is a special node of Spin type. Consequently, the value of the node $k$ must be $k_k = 4$.

- $G_k = SU(2P+1)$. In this case $Z_k$ does not have a $\mathbb{Z}_2$ element, but $\pi_k(\alpha)$ must be a $\mathbb{Z}_2$ element in $Z_k$. This contradicts the existence of 2-group symmetry.

- $G_k = SU(2P)$. Assume that there is no other non-flavor node connected to $j$. Then there is a flavor node $a$ neighboring $j$ which carries $F_a = \mathrm{Spin}(4Q+2)$. The string associated to $j$ provides states charged as $\mathbf{S} \otimes \mathbf{1} \otimes \mathbf{S}, \mathbf{1} \otimes \mathbf{\Lambda}^2 \otimes \mathbf{1}, \mathbf{S} \otimes \mathbf{F} \otimes \mathbf{C}$ of $G_i \times G_k \times F_a$. From the

first string state, we see that $\pi_a(\beta)$ is the $\mathbb{Z}_2$ element of $Z_a$, which is not allowed since $\beta$ is a part of 1-form symmetry, and hence cannot involve non-trivial elements of flavor centers.

Now assume there is another non-flavor node $l$ connected to $j$. The value of $l$ must be 4, and hence $l$ must carry $G_l = \mathrm{Spin}(q)$. The configuration formed by the four nodes $i, j, k, l$ is a LST, so no more non-flavor nodes can be added to it. Implementing the constraints on the ranks, we find that such a model cannot have any flavor symmetry, and hence cannot carry 2-group symmetry.

- Consider $G_k = \mathrm{Spin}(4P)$ with a bifundamental half-hyper between $j$ and $k$. Suppose first that there is no other non-flavor neighbor $l$ of $j$. There is a string state charged as a bi-spinor of $G_i \times G_k$. For this state to be left invariant under $\beta$, $\pi_k(\beta)$ must be an order two element in $Z_k = \mathbb{Z}_2^2$. However, if this is true then $\pi_k(\alpha)$ must be a $\mathbb{Z}_4$ element inside $Z_k$, which is a contradiction.

  Now assume that there is a non-flavor neighbor $l$ of $j$. If $l$ is non-gauge-theoretic, it must carry $G_l = SU(1)$, constraining $G_j$ to be $Sp(1)$. Moreover, it induces a trapped half-hyper transforming as $F$ of $G_j$ at the intersection of $l$ and $j$, which does not transform under any flavor symmetry. This $\frac{1}{2}F$ transforms under $\alpha$ and destroys the 2-group symmetry. Thus $l$ must be gauge-theoretic. We must have $G_l = \mathrm{Spin}(4q+2)$. There might be an additional flavor node $a$ attached to $j$ carrying $F_a = \mathrm{Spin}(4r)$. Then, we have string states transforming as $S \otimes S \otimes S \otimes S, S \otimes S \otimes C \otimes C, S \otimes C \otimes S \otimes C, S \otimes C \otimes C \otimes S$ under $\mathcal{G}_i \times \mathcal{G}_l \times \mathcal{G}_k \times F_a$. Along with hypers, one can see that this is consistent with the existence of 2-group symmetry. $l$ is a special node of Spin type, and hence its value must be $k_l = 4$. We can also consider adding spinor and cospinor matter for $G_k$. This is possible only for $G_k = \mathrm{Spin}(8)$ or $\mathrm{Spin}(12)$. Implementing the rank constraints, we find only the following possibility:

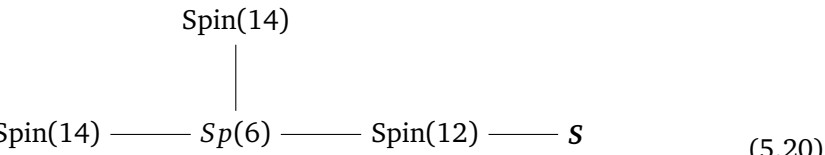

$$\qquad (5.20)$$

  in which the $U(1)$ symmetry rotating the spinor participates in $\alpha$. This means that the associated Postnikov class is trivial as we discussed earlier in this section.

- Consider $G_k = \mathrm{Spin}(2P+1)$ with a bifundamental half-hyper between $j$ and $k$. $\pi_j(\alpha)$ acts on this bifundamental, so has to compensated by an element from $Z_k$. However, no element of $Z_k$ acts non-trivially on the bifundamental. Thus, such a node is not allowed.

- For $G_k = \mathrm{Spin}(7)$, we can have a half-hyper in $F \otimes S$ of $G_j \times G_k$. Assume first that there is no other non-flavor node $l$ connected to $j$. In this case, we have a string state transforming as $S \otimes 1$ of $G_i \times G_k$ which transforms under $\beta$, leading to a contradiction.

  Now assume that there is a non-flavor node $l$. The value of $l$ must be at least 3. Thus, $l$ must be gauge-theoretic. $G_l$ cannot be $SU(3)$ since $n > 0$, so we must have $G_l = \mathrm{Spin}(2P)$ with a half-bifundamental between $l$ and $j$. Assume $P$ is even. Then we must have a flavor node $a$ neighboring $j$ and carrying $F_a = \mathrm{Spin}(4Q+2)$. Moreover, we have a string state transforming as $S \otimes 1 \otimes S \otimes S$ of $G_i \times G_k \times G_l \times F_a$. For this string state to be left invariant by $\alpha$, we must have $\pi_a(\alpha)$ as an order four element of $Z_a$. This implies that $\pi_a(\beta)$ is an order two element of $Z_a$, which is in contradiction with the fact that $\beta$ is part of 1-form symmetry.

  Now consider the case $P$ odd. In this case, rank constraints imply that there is no possible model.

- $G_k = G_2$ is not allowed since $Z_k$ must contain $\pi_k(\alpha)$ as a $\mathbb{Z}_2$ element but $Z_k = \mathbb{Z}_1$.

Another possibility for $j$ is $G_j = SU(N)$. For this case, $N$ must be even, since $\pi_j(\alpha)$ has to be the $\mathbb{Z}_2$ element of $Z_j$. Let $\mathcal{J}$ be the set of nodes $k$ such that $\mathcal{G}_k = SU(P)$ and $k$ is connected to $j$ by a chain of $SU$ nodes.

$$\boxed{G_i = \mathrm{Spin}(4M+2)} \longrightarrow \boxed{G_j = SU(N)} \dashrightarrow \boxed{\mathcal{J} \ni G_k = SU(P)} \longrightarrow \boxed{G_l = ?} \quad . \tag{5.21}$$

We now study gauge-theoretic neighbors $l$ of $\mathcal{J}$:

- We cannot have a Spin neighbor of $\mathcal{J}$ by the rank constraints.

- We cannot have a $G_l = G_2$ neighbor since we would need $\pi_l(\alpha)$ to be a $\mathbb{Z}_2$ element in $Z_l = \mathbb{Z}_1$, which is not possible.

- Consider $G_l = \mathrm{Spin}(7)$ and let it be a neighbor of $k \in \mathcal{J}$ with $G_k = SU(2)$. In this case there is a half-hyper in $\boldsymbol{F} \otimes \boldsymbol{S}$ of $G_k \times G_l$. This is not allowed since all the matter content associated to $k$ is gauged by $l$, and it is therefore not possible to connect $k$ to $i$.

- Consider $G_l = Sp(Q)$. The rank constraints imply that the only possibility is

$$\mathrm{Spin}(4M+2) \longrightarrow SU(4M-6) \longrightarrow SU(4M-14) \dashrightarrow SU(2Q+8) \longrightarrow Sp(Q) \quad , \tag{5.22}$$

  which has no flavor symmetry and hence no 2-group symmetry.

To finish the analysis of possible gauge theory nodes surrounding $i$, we need to understand possible neighbors $l$ of $k$ carrying $G_k = \mathrm{Spin}(4p)$ with value $k_k = 4$, which arises in a sub-graph of the form:

$$\boxed{G_i = \mathrm{Spin}(4M+2)} \longrightarrow \boxed{G_j = Sp(N)} \longrightarrow \boxed{G_k = \mathrm{Spin}(4P)} \longrightarrow \boxed{G_l = ?}$$
$$\boxed{\mathrm{Spin}(4Q+2)} \qquad\qquad\qquad . \tag{5.23}$$

First consider the case $G_l = Sp(R)$. Suppose there are no other non-flavor neighbors of $l$. We can have a flavor node $a$ attached to $l$ carrying $F_a = \mathrm{Spin}(4S)$. The string states arising from $l$ transform as $\boldsymbol{S} \otimes \boldsymbol{S}, \boldsymbol{C} \otimes \boldsymbol{C}$ of $G_k \times F_a$. From this one can see that the existence of such a node $l$ is consistent with 2-group symmetry. Let us consider possible neighbors $h$ of $l$:

- We can have $G_h = \mathrm{Spin}(4S)$ with $k_h = 4$ consistently. For $k_h < 4$, it is not possible to satisfy rank constraints.

- We cannot have $G_h = \mathrm{Spin}(4S+2)$ as then there must be a non-trivial flavor symmetry node $a$ attached to $l$ carrying $F_a = \mathrm{Spin}(4T+2)$ with the property that $\pi_a(\beta)$ is $\mathbb{Z}_2$ element of $F_a$, which is a contradiction with the fact that $\beta$ is a part of 1-from symmetry.

- $G_h = \mathrm{Spin}(2S+1)$ with bifundamental half-hyper between $h$ and $l$ is not allowed as one needs $\pi_h(\alpha)$ to act non-trivially on the vector rep of $G_h$, but no element of $Z_h$ acts non-trivially on the vector rep.

- $G_h = \mathrm{Spin}(7)$ with half-hyper in $\boldsymbol{F} \otimes \boldsymbol{S}$ of $\mathcal{G}_l \times \mathcal{G}_h$ is not allowed by rank constraints.

- $G_h = G_2$ or $SU(2S+1)$ are not allowed since we would need $\pi_h(\alpha)$ to be a $\mathbb{Z}_2$ element in $Z_h$, which does not exist.

- Choosing $G_h = SU(2S)$ constrains the model fully via rank constraints to be

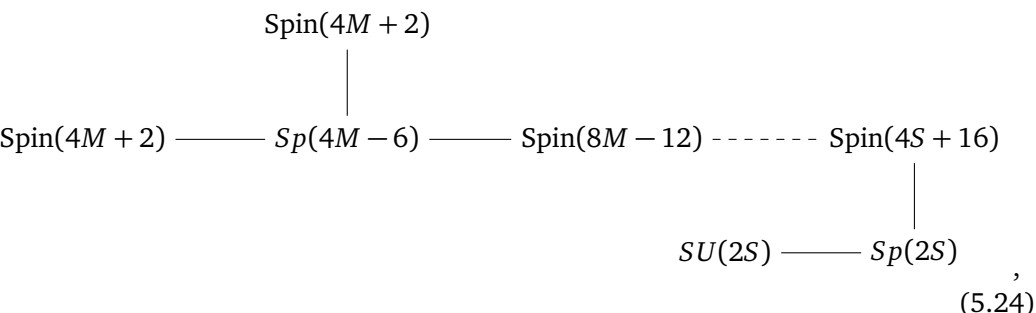

$$\tag{5.24}$$

  which does not have any flavor symmetry, and hence no 2-group symmetry.

Another possibility is $G_l = SU(R)$ which can be rejected because it does not satisfy the rank constraints.

We now explore possible non-gauge-theoretic nodes $l$ neighboring the above studied gauge-theoretic block surrounding the special node $i$ of Spin type:

- For $G_l = SU(1)$, it traps a $\frac{1}{2}\boldsymbol{F}$ of the neighboring node $k$ carrying $SU(2)$ or $Sp(1)$. $\pi_k(\alpha)$ acts on this $\frac{1}{2}\boldsymbol{F}$ and this action cannot be compensated by any flavor symmetry. So $G_l = SU(1)$ is not allowed.

- Consider $G_l = Sp(0)$ and its neighbor $G_k = SU(P \leq 9)$. This is not consistent with the rank constraints.

- Consider $G_l = Sp(0)$ and its neighbor $G_k = \mathrm{Spin}(4P+2)$. We have two possibilities. For $G_k = \mathrm{Spin}(14)$, the flavor symmetry attached to $G_l$ is only $\mathfrak{u}(1)$. So, no other node can be attached to $l$. The node $l$ contributes a string state transforming as $\boldsymbol{S}$ of $\mathrm{Spin}(14)$ which breaks the 1-form symmetry. For $G_k = \mathrm{Spin}(10)$, we must have a node $h$ neighboring $l$ and carrying $G_h = SU(4)$. Moreover, $\pi_h(\alpha)$ and $\pi_h(\beta)$ are $\mathbb{Z}_4$ and $\mathbb{Z}_2$ elements inside $Z_h$ respectively. This implies that $h$ is a special node of $SU$ type, but we have already ruled out the existence of such a special node in the previous subsection.

- Consider $G_l = Sp(0)$ and its neighbor $G_k = \mathrm{Spin}(4P)$. This is only possible for $P \leq 4$. However, the rank constraints force $P \geq 5$, leading to a contradiction.

Thus, no such non-gauge-theoretic nodes are allowed.

Implementing the above discussed constraints on the possible theories having special nodes of Spin type, we are lead to the building blocks listed in section 5.1. Combining these building blocks results in the classification presented in section 3.3.

# 6 Conclusions

This paper uncovers the existence of global 2-group symmetries in 6d SCFTs and provides a complete classification of such theories carrying 2-group symmetries of the type discussed here. These are 2-groups formed out of *discrete* 1-form symmetries and continuous 0-form flavor symmetry groups. The result is perhaps surprising in view of the no-go theorems for 2-groups having continuous 1-form symmetries and continuous 0-form symmetries from general principles [15].

Moreover the reasoning put forward to check for the existence of 2-groups is applicable quite generally, in all dimensions ($d = 3, 4, 5, 6$ specifically), and the analysis in section 2 is completely general. The only dimension-dependent features arise in the type of charged objects one needs to incorporate (of course matter multiplets, but also non-perturbative states that can be dimension specific). The simplest avatar of this universally present 2-group symmetry can be seen in gauge theories Spin($4N + 2$) with $N_F$ vectors with 8 supercharges. These theories have 2-groups in all dimensions; in $\mathcal{N} = 1$ and $\mathcal{N} = 0$ (non-supersymmetric) theories in 4d this was observed in [14, 23]. It is in view of this generality that we provide a mathematica code `TwoGroupCalculator.nb`, which can be used in this general setting to determine the 2-group symmetries of a given quiver gauge theory in $d$ dimensions.

Thanks to the classification of 6d SCFTs we are able to determine all possible theories with 2-groups of the type studied in this paper, which are summarized in table 1. Their tensor branches are quivers built out of $\mathfrak{so} - \mathfrak{sp}$ or $\mathfrak{so} - \mathfrak{su}$ gauge algebras. We also show that LSTs do not have these types of 2-groups (formed from discrete 1-form symmetries and continuous flavor groups), though unlike 6d SCFTs they instead do have continuous 2-groups. In our analysis it is crucial to determine the global form of the flavor symmetry group $\mathcal{F}$ of the 6d theory. This is part of our analysis and can be extracted from the computation of $\mathcal{E}$ and its projection $\mathcal{Z}$ onto the center of the flavor symmetry $Z_F$. We also discussed that in the case of abelian flavor symmetry factors there can be ABJ anomalies that break these symmetries to discrete subgroups.

Global symmetries can have 't Hooft anomalies and we identified two such anomalies in 6d: one is the standard "dual" mixed anomaly for a 2-group, in this case between the 0-form and 3-form symmetry (which is the symmetry obtained after gauging the 1-form symmetry that participates in the 2-group). The other 't Hooft anomaly is a mixed anomaly between the 0-form and 1-form symmetry, which is similar to known anomalies in 4d and 5d.

Clearly, the study of generalized symmetries, and in particular higher-group and categorical symmetries are at a starting point. It would be exciting to explore the physical implications of higher-groups, similar to the relevance of higher-form symmetries (e.g. for confinement). The higher-groups can have 't Hooft anomalies, in addition to the anomalies we have discussed here. It would be interesting to derive these from first principles from the F-theory geometric realization, and to explore their potential implications for the UV fixed points.

## Acknowledgements

We thank Mathew Bullimore, Andrea Ferrari and Craig Lawrie for discussions. The work is supported by the European Union's Horizon 2020 Framework: ERC grants 682608 (FA, LB, and SSN) and 787185 (LB). SSN is also supported in part by the "Simons Collaboration on Special Holonomy in Geometry, Analysis and Physics".

## A  Mathematica **Code for 2-Groups:** `TwoGroupCalculator.nb`

We also provide a supplementary notebook `TwoGroupCalculator.nb`. With this at hand the reader can interrogate examples of their own. Specifically, given the gauge and flavour centers and hyper/string charges, the notebook allows its user to calculate the 1-form symmetry $\Gamma^{(1)}$, $\mathcal{E}$, and $\mathcal{Z}$. Furthermore it calculates the mappings that define the sequence

$$0 \to \Gamma^{(1)} \to \mathcal{E} \to \mathcal{Z} \to 0 \,. \tag{A.1}$$

Take, for example, a type 5 quiver:

$$
\begin{array}{ccc}
\mathfrak{su}(12) & \underline{\phantom{XXXXX}} & \mathfrak{so}(22) \\
2 & & 2 \\
| & & | \\
[\mathfrak{u}(2)] & & [\mathfrak{sp}(2)]
\end{array}
\tag{A.2}
$$

The required input for the code is a list of gauge centers ($\mathbb{Z}_{12} \times \mathbb{Z}_4$)

$$
\{12, 4\}, \tag{A.3}
$$

and flavour centers ($\mathbb{Z}_2 \times \mathbb{Z}_2$)

$$
\{2, 2\}, \tag{A.4}
$$

and matter charges as as list of associations

$$
\{< |1 \to 1, 3 \to 1| >, < |1 \to 1, 2 \to 2| >, < |2 \to 2, 4 \to 1| >\}. \tag{A.5}
$$

with the notation $< |\text{node}_i \to \text{charge}_j| >$ representing the charge of a given hyper under node $i$, where $i$ is a numerical label for each node. E.g. $< |1 \to 1, 3 \to 1| >$ means that the first hyper has charge 1 under node 1 ($\mathfrak{su}(12)$) and charge 1 under node 3 ($\mathfrak{u}(2)$).

Included in the notebook is an explicit example for each type of 6d quiver, as well as detailed worked examples to explain how to calculate desired attributes of any quiver.

## B  Detailed Quiver Example

In this appendix we will discuss in depth the 2-group for the 6d SCFT with tensor branch

$$
\begin{array}{c}
[\mathfrak{sp}(N-3)] \\
| \\
\mathfrak{so}(4N+2) \\
4 \\
| \\
[\mathfrak{sp}(N-5)] - \underset{4}{\mathfrak{so}(4N)} \underline{\phantom{XX}} \underset{1}{\mathfrak{sp}(3N-3)} \underline{\phantom{XX}} \underset{4}{\mathfrak{so}(4N+2)} - [\mathfrak{sp}(N-3)]
\end{array}
\tag{B.1}
$$

with simply-connected gauge and flavour symmetry groups

$$
\begin{aligned}
G &= \mathrm{Spin}(4N) \times \mathrm{Spin}(4N+2)^2 \times Sp(3N-3), \\
F &= Sp(N-3)^2 \times Sp(N-5).
\end{aligned}
\tag{B.2}
$$

A local consistency condition is that the flavours attached to the $\mathfrak{sp}(3N-3)$ node are soaked up by the surrounding nodes

$$
2(3N-3) + 8 = \frac{1}{2}(4N+2+4N+2+4N). \tag{B.3}
$$

We can write the matter content in terms of representations of $(\mathfrak{so}(4N), \mathfrak{sp}(3N-3), \mathfrak{so}(4N+2), \mathfrak{so}(4N+2))$ as

$$
\begin{aligned}
(N-5)(\boldsymbol{F}, 1, 1, 1) &\oplus \frac{1}{2}(\boldsymbol{F}, \boldsymbol{F}, 1, 1) \oplus \frac{1}{2}(1, \boldsymbol{F}, \boldsymbol{F}, 1) \oplus \frac{1}{2}(1, \boldsymbol{F}, 1, \boldsymbol{F}) \\
&\oplus (N-3)(1, 1, 1, \boldsymbol{F}) \oplus (N-3)(1, 1, \boldsymbol{F}, 1),
\end{aligned}
\tag{B.4}
$$

The hypermultiplet content described above can also be written in terms of charges under the center symmetries (table 3).

$$Z_G = (\mathbb{Z}_2 \times \mathbb{Z}_2) \times \mathbb{Z}_4 \times \mathbb{Z}_2 \times \mathbb{Z}_4,$$
$$Z_F = \mathbb{Z}_2 \times \mathbb{Z}_2 \times \mathbb{Z}_2.$$

(B.5)

Here we employ notation $(a, b, c)$ to represent the charges under the flavour symmetries running anti-clockwise around the $Sp$ flavour nodes starting from the bottom left. We must also be

Table 3: Matter content in terms of $Z_G \times Z_F$ charges.

| Hypermultiplet | $Z_G$ charge | $Z_F$ charge |
|---|---|---|
| $(\boldsymbol{F}, 1, 1, 1)$ | ((1 mod 2,1 mod 2),0,0,0) | (1 mod 2,0,0) |
| $\frac{1}{2}(\boldsymbol{F}, \boldsymbol{F}, 1, 1)$ | ((1 mod 2,1 mod 2),1 mod 2,0,0) | (0,0,0) |
| $\frac{1}{2}(1, \boldsymbol{F}, \boldsymbol{F}, 1)$ | ((0,0),1 mod 2,2 mod 4,0) | (0,0,0) |
| $\frac{1}{2}(1, \boldsymbol{F}, 1, \boldsymbol{F})$ | ((0,0),1 mod 2,0,2 mod 4) | (0,0,0) |
| $(1, 1, 1, \boldsymbol{F})$ | ((0,0),0,0,2 mod 4) | (0,1 mod 2,0) |
| $(1, 1, \boldsymbol{F}, 1)$ | ((0,0),0,2 mod 4,0) | (0,0,1 mod 2) |

careful about charged strings. In this case, the string charges under $\mathrm{Spin}(4N) \times \mathrm{Spin}(4N+2)^2$ gauge centers are given in table 4 .

Table 4: String content in terms of $Z_G \times Z_F$ charges.

| Charged Strings | $Z_G$ charge | $Z_F$ charge |
|---|---|---|
| String 1 | ((1 mod 2, 0 mod 2),0,1 mod 4, 1 mod 4) | (0,0,0) |
| String 2 | ((1 mod 2, 0 mod 2),0,3 mod 4, 3 mod 4) | (0,0,0) |
| String 3 | ((0 mod 2, 1 mod 2),0,1 mod 4, 3 mod 4) | (0,0,0) |
| String 4 | ((0 mod 2, 1 mod 2),0,3 mod 4, 1 mod 4) | (0,0,0) |

With this in place we can ask: what is the maximal subgroup of $Z_G \times Z_F$ that leaves these charged states invariant? Neglecting the flavour charges, this calculation would give us the 1-form symmetry of the theory (the subgroup of the gauge centers acting trivially on the matter and strings). Including them, we obtain $\mathcal{E}$ required for the definition of the structure group.

**Calculating the 1-form symmetry**

The task is now, in principle, simple. Take, for example, the generator of one of the $\mathbb{Z}_4$ gauge centers

$$\langle (0,0), 0, \frac{1}{4}, 0 \rangle.$$

(B.6)

We can see that this is immediately reduced to a $\mathbb{Z}_2$ subgroup by the matter $(1, 1, \boldsymbol{F}, 1)$

$$\langle (0,0), 0, \frac{1}{4}, 0 \rangle \cdot ((0,0), 0, 2 \bmod 4, 0) \neq 0 \bmod \mathbb{Z}.$$

(B.7)

Notice that the matter can also reduce to a mixed subgroup of two gauge factors. For example, the matter $\frac{1}{2}(1, \boldsymbol{F}, \boldsymbol{F}, 1)$ reduces a combined $\mathbb{Z}_2 \times \mathbb{Z}_4$ to a diagonal $\mathbb{Z}_2$ subgroup

$$\langle (0,0), \frac{1}{2}, \frac{1}{4}, 0 \rangle \cdot ((0,0), 1, 2 \bmod 4, 0) \neq 0 \bmod \mathbb{Z}.$$
$$\to \langle (0,0), \frac{1}{2}, \frac{2}{4}, 0 \rangle \cdot ((0,0), 1, 2 \bmod 4, 0) = 0 \bmod \mathbb{Z}.$$

(B.8)

Playing this game throughout, we obtain the 1-form symmetry group $\Gamma^{(1)} = \mathbb{Z}_2 \times \mathbb{Z}_2$ and its generators

$$\gamma_1 = \langle (\frac{1}{2}, \frac{1}{2}), 0, \frac{2}{4}, 0 \rangle, \quad \gamma_2 = \langle (\frac{1}{2}, \frac{1}{2}), 0, 0, \frac{2}{4} \rangle. \tag{B.9}$$

**Calculating $\mathcal{E}$ and the embedding of $\Gamma^{(1)}$**

In this example we can identify

$$\Gamma^{(1)} = \mathbb{Z}_2 \times \mathbb{Z}_2, \quad \mathcal{E} = \mathbb{Z}_2 \times \mathbb{Z}_4. \tag{B.10}$$

The non-trivial question is: how is $\Gamma^{(1)}$ embedded in $\mathcal{E}$? We can use the explicit charge presentation to determine that a diagonal $\mathbb{Z}_2$ combination of the $\Gamma^{(1)}$ generators is enhanced inside the $\mathbb{Z}_4$. The generator of the $\mathbb{Z}_4 \subset \mathcal{E}$ is (under $Z_G \times Z_F$)

$$\alpha = \langle (\frac{1}{2}, 0), \frac{1}{2}, \frac{1}{4}, \frac{1}{4} | \frac{1}{2}, \frac{1}{2}, \frac{1}{2} \rangle. \tag{B.11}$$

We notice that $\alpha^2|_G = \gamma_1 \cdot \gamma_2$: explicit confirmation that a diagonal (in our chosen basis of generators) $\mathbb{Z}_2$ is enhanced.

# C    Chern Classes, Characters, and Integrality

Let us recall some basic identities about Chern classes. The Chern forms for a complex vector bundle are defined by the following expansion,

$$\begin{aligned}
\sum_j c_j(V) t^j =& \mathbb{I}_{d_\rho} t + i \mathrm{tr}_\rho(F) - \frac{1}{2} \left( \mathrm{tr}_\rho(F^2) - \mathrm{tr}_\rho(F)^2 \right) t^2 \\
& + i \frac{1}{6} \left( -2 \mathrm{tr}_\rho(F^2)^3 + 3 \mathrm{tr}_\rho(F^2) \mathrm{tr}_\rho(F) - \mathrm{tr}_\rho(F)^6 \right) t^3 + \dots,
\end{aligned} \tag{C.1}$$

where $d_\rho$ is the dimension of the representation. The Chern character for a vector bundle given by a representation of a Lie algebra $\rho(\mathfrak{g})$ is instead defined as,

$$\begin{aligned}
\mathrm{ch}(V) = \mathrm{tr}_\rho \left( \exp(iF) \right) &= 1 + i \mathrm{tr}_\rho(F) t - \frac{\mathrm{tr}_\rho(F^2)}{2} + i \frac{\mathrm{tr}_\rho(F^2)^3}{6} \dots \\
&= d_\rho + c_1(V) + \frac{1}{2} \left( c_1(V)^2 - 2c_2(V) \right) + \frac{1}{6} \left( 3c_2(V) - 3c_1(V)c_2(V) + c_1(V)^3 \right) + \dots.
\end{aligned} \tag{C.2}$$

For the abelian vector bundle we have,

$$\mathrm{ch}(U(1)_q) = 1 + q c_1(V) + \frac{q^2 c_1(V)^2}{2} + \frac{q^3 c_1(V)^3}{6} + \dots. \tag{C.3}$$

If we have $W = V \otimes U$ where these are all vector bundles, we can decompose the curvatures and Chern classes using the following formula,

$$\mathrm{ch}(W) = \mathrm{ch}(V) \mathrm{ch}(U). \tag{C.4}$$

In particular let us decompose the vector bundle where the fundamental representation of $\mathfrak{u}(N)$ acts, and with an abuse of notation like in (C.3), we will look at each term in the expansion

of the following formula $\mathrm{ch}(\mathfrak{u}(N)) = \mathrm{ch}(\mathfrak{su}(N))\mathrm{ch}(\mathfrak{u}(1))$. By using (C.2) and (C.3), with $q = 1$ and $d_\rho = N$ we obtain

$$
\begin{aligned}
c_1(\mathfrak{u}(N)) &= N c_1(\mathfrak{u}(1)), \\
c_2(\mathfrak{u}(N)) &= c_2(\mathfrak{su}(N)) + \frac{N(N-1)}{2} c_1(\mathfrak{u}(1))^2, \\
c_3(\mathfrak{u}(N)) &= c_3(\mathfrak{su}(N)) + (N-2)c_2(\mathfrak{su}(N))c_1(\mathfrak{u}(1)) + \frac{N(N-1)(N-2)}{6} c_1(\mathfrak{u}(1))^3.
\end{aligned}
\tag{C.5}
$$

We would like to discuss the integrality of the curvature and Chern classes when integrated on a 6-manifold with an almost complex structure or submanifold thereof. In particular, a very important formula which is a corollary of the Atiyah-Singer theorem is that the index of a vector bundle reads

$$
\mathrm{Ind}(V) = \sum_p (-1)^p h^p(M_6, V) = \int_{M_6} \mathrm{ch}(V)\mathrm{td}(M_6)|_6 \quad \in \quad \mathbb{Z},
\tag{C.6}
$$

where $h^p$ indicates the dimension of various integral cohomologies, and the Todd class is defined as follows

$$
\mathrm{td}(M_6) = 1 + \frac{1}{2}c_1 + \frac{1}{12}(c_1^2 + c_2) + \frac{c_1 c_2}{24} + \dots,
\tag{C.7}
$$

where when there is no argument for the Chern classes we mean the one of the manifold under inspection. For the vector bundle where the fundamental representation of $\mathfrak{su}(N)$ acts, we have that

$$
\mathrm{Ind}(V_{\mathbf{fund}(\mathfrak{su}(N))}) = \frac{c_3(\mathfrak{su}(N))}{2} + \frac{1}{2}c_1 c_2(\mathfrak{su}(N)) + \frac{c_1 c_2}{24}.
\tag{C.8}
$$

In addition on a manifold with an almost complex structure (or equivalently with a spin$^c$ structure) we have that $\int_\Sigma c_1 = 2\mathrm{genus}(\Sigma) - 2 \in 2\mathbb{Z}$ and $c_1 c_2 \in 24\mathbb{Z}$, [93]. This together with the integrality of the index implies that $c_3(\mathfrak{su}(N)) \in 2\mathbb{Z}$.

We can use this to prove also that $c_3(\mathfrak{u}(2n)) \in 2\mathbb{Z}$. Specializing the third equation of (C.5) to $N = 2n$ we get,

$$
c_3(\mathfrak{u}(2n)) = c_3(\mathfrak{su}(2n)) + 2(n-1)c_2(\mathfrak{su}(2n))c_1(\mathfrak{u}(1)) + \frac{2n(2n-1)(2n-2)}{6} c_1(\mathfrak{u}(1))^3, \tag{C.9}
$$

where all the terms on the right hand side are in $2\mathbb{Z}$.

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
