# Peer review of "-Group Symmetries and their Classification in 6d"

_SciPost Physics, doi:SciPost Phys. 12, 098 (2022)_

## Round 1 · Referee Report · Anonymous (Referee 1) · 2021-12-13

Report
This article studied two-group symmetries mainly in six dimensional SCFTs. One of the important results is the classification of 6d SCFTs that exhibit two-group symmetries. The authors showed that they come in seven types. They also provided an algorithm and a Mathematica file that computed the extension of the one-form symmetry and the center symmetry as well as the flavor symmetry group. The method was also demonstrated clearly using explicit examples.
The referee has one question that would like the authors to address prior to publication. As pointed out by the authors, six out of seven classes of 6d SCFTs that are consistent with the two-group symmetry involve the frozen phase of F-theory. Are there any meaningful or physical relations between the non-trivial two-group symmetries and such frozen phases? If so, I would like to ask the authors to elucidate this point further in the article.
After this minor point being addressed, the referee recommends this article to be published in SciPost.
The referee has one question that would like the authors to address prior to publication. As pointed out by the authors, six out of seven classes of 6d SCFTs that are consistent with the two-group symmetry involve the frozen phase of F-theory. Are there any meaningful or physical relations between the non-trivial two-group symmetries and such frozen phases? If so, I would like to ask the authors to elucidate this point further in the article.
After this minor point being addressed, the referee recommends this article to be published in SciPost.

Author: Dewi Gould on 2021-12-15 [id 2029]
(in reply to Report 1 on 2021-12-13)We thank the referees for their insightful comments. We wish to respond to the question regarding any possible link between the 2-group symmetries found in this work, and the frozen phase of F-theory in which several of our examples reside. The central aim of this work, as highlighted by the referee, was to identify the existence of non-trivial 2-group symmetries in 6d SCFTs, and provide a classification of theories which exhibit such symmetries. The referee correctly points out that six out of seven classes of such theories arise in the frozen phase of F theory.
To the best of our knowledge, any physical relations/ consequences identified are general properties and do not rely on the frozen/ un-frozen origins of the given theory. We do not believe that there is any strong relation deriving from 2-group symmetries that is unique to the frozen phase. It is definitely of interest to explore the frozen phase of F-theory further, but we believe this broader avenue of research is beyond the scope of our current work.

---

## Round 1 · Referee Report · Anonymous (Referee 2) · 2021-12-14

Report
The paper classified 6d SCFTs with a particular 2-group symmetry that combines the 0-form flavor symmetry and discrete 1-form symmetry. The Postnikov class that specifies the 2-group is the Bockstein of the extension class for the universal covering group of the flavor symmetry. While such 2-group symmetry appears in many examples in lower spacetime dimensions, it is nice to see the classification of 6d SCFTs using such symmetry. The authors also discussed a new mixed anomaly between the flavor symmetry and the 1-form symmetry. The methods are demonstrated using explicit examples.

---

## Round 2 · List of Changes

footnote added to address minor comment in review process.

---

## Editorial Decision

published